# Inter-site structural heterogeneity induction of single atom Fe catalysts for robust oxygen reduction

Peng Zhang[1,6], Hsiao-Chien Chen[2,3,6], Houyu Zhu[4,6], Kuo Chen[1], Tuya Li[4], Yilin Zhao[1], Jiaye Li[1], Ruanbo Hu[5], Siying Huang[1], Wei Zhu [5], Yunqi Liu[1] ✉ & Yuan Pan [1] ✉

Metal-nitrogen-carbon catalysts with hierarchically dispersed porosity are deemed as efficient geometry for oxygen reduction reaction (ORR). However, catalytic performance determined by individual and interacting sites originating from structural heterogeneity is particularly elusive and yet remains to be understood. Here, an efficient hierarchically porous Fe single atom catalyst (Fe SAs-HP) is prepared with Fe atoms densely resided at micropores and mesopores. Fe SAs-HP exhibits robust ORR performance with half-wave potential of 0.94 V and turnover frequency of 5.99 $e^{-1}s^{-1}site^{-1}$ at 0.80 V. Theoretical simulations unravel a structural heterogeneity induced optimization, where mesoporous $Fe-N_4$ acts as real active centers as a result of long-range electron regulation by adjacent microporous sites, facilitating $O_2$ activation and desorption of key intermediate *OH. Multilevel *operando* characterization results identify active Fe sites undergo a dynamic evolution from basic $Fe-N_4$ to active $Fe-N_3$ under working conditions. Our findings reveal the structural origin of enhanced intrinsic activity for hierarchically porous $Fe-N_4$ sites.

Ever-growing concerns about energy and environmental issues call for clean energy. Fuel cells and metal-air batteries represent promising candidates for the next generation of energy conversion and storage system due to their high energy density, low cost, eco-friendliness, etc[1,2]. The development of such advanced energy conversion configurations, however, is hampered by sluggish oxygen reduction reaction (ORR) kinetics and the high cost of precious metal catalysts at air cathodes. The emergency of single-atom catalysts (SACs) with high atom utilization and tunable active sites has reinvigorated intense attention for addressing activity, durability, and high costs of noble metal-based catalysts[3–5]. Among various SACs, $Fe-N_4$ sites with well-defined porphyrin-like structures are generally considered the most active and selective elelctrocatalysts for ORR. However, $O_2$ adsorption

and activation were impeded due to symmetric electron distribution of well-defined $Fe-N_4$ structure[6]. Great endeavors have been devoted to exploiting efficient Fe SACs with asymmetric coordination structure, such as edge-type $Fe-N_4$, $Fe-N_3S_1$, $Fe-N_4P_1$, etc[7,8]. Despite great progress being made so far, it was soon discovered the performance of batteries was beyond the electrocatalyst matters. Since the ORR took place at a triple-phase interface, the geometric structure of electrocatalysts will also greatly affect electron and mass transport[9,10]. Generally, mesoporous structure was considered to guarantee a fast mass transport, and microporous structure promoted faster shutting of electrons and contributed to resisting the intrusion of water during ORR[11,12]. Thereafter, SACs with hierarchically porous structure were regarded as an efficient geometry for ORR. While several studies have

[1]State Key Laboratory of Heavy Oil Processing, China University of Petroleum (East China), Qingdao 266580, China. [2]Center for Reliability Science and Technologies, Chang Gung University, Taoyuan 33302, Taiwan. [3]Kidney Research Center, Department of Nephrology, Chang Gung Memorial Hospital, Linkou, Taoyuan 33305, Taiwan. [4]School of Materials Science and Engineering, China University of Petroleum (East China), Qingdao 266580, China. [5]State Key Lab of Organic-Inorganic Composites, Beijing University of Chemical Technology, Beijing 100029, China. [6]These authors contributed equally: Peng Zhang, Hsiao-Chien Chen, Houyu Zhu. ✉e-mail: liuyq@upc.edu.cn; panyuan@upc.edu.cn

been carried out to construct efficient SACs with hierarchical porous structure, little work has established a relationship between individual intrinsic activity and structure due in part to a mass transport-controlled process provided by rotation disk electrode. Additionally, the rational design of efficient SACs for ORR also relies heavily on basic understanding of catalytic under working conditions at atomic precise. In general, the dynamic switching behavior of SACs was sensitive to coordinative environment, even for the same central metal-based SACs[13,14]. For instance, Fe-N-C moiety in the plane of carbon–nitrogen matrix might undergo three dynamic modes by monitoring the central Fe atoms moving towards or away from N-4 plane in relation to the pyrolysis temperature and geometric structure of Fe-N$_4$ sites[15]. The dynamic structure was found to essentially govern ORR activities. Therefore, an extensive comprehension of the relationship between structure and intrinsic activity, as well as identifying dynamic evolution of active centers, is highly desirable but remained a great challenge.

Here we demonstrate a comprehensive understanding of hierarchically porous Fe-N$_4$ sites for ORR in combination with experimental and theoretical methods. By engineering the pore structure of Fe-N$_4$ sites, it allowed one to determine catalytic behavior of individual sites along with structural heterogeneity induced effect. Coal tar pitch (CTP), composed of polycyclic aromatic hydrocarbons (PAH), was regarded as high-quality carbon resource and has been widely used in the field of energy storage and conversion systems due to its high conductivity and structure flexibility[16–18]. Herein, an encapsulation-pyrolysis-evaporation strategy was developed to prepare hierarchically porous Fe SACs (Fe SAs-HP) immobilized on CTP-derived carbon substrates. Naturally, heme chloride was carefully selected as Fe source due to strong π-π interactions between heme chloride macrocycle and PAH, which was conducive to evenly dispersing Fe atoms into carbon networks and avoiding the agglomeration of undesirable metallic iron nanoparticles. ZnO, sacrificial templates, and NH$_3$ atmosphere were allowed to react with carbon substrate under high temperatures to form abundant mesoporous edges and micropores to immobilize Fe atoms, forming active Fe moieties.

Hierarchically porous Fe SAs-HP exhibited remarkable ORR activity with half-wave potential ($E_{1/2}$) of 0.94 V and excellent long-term durability for 30 k cyclic voltammetry (CV) cycles in alkaline media. In addition, the superior catalytic performance of Fe SAs-HP was also witnessed by the high mass activity (MA) ($4.14 \times 10^4$ A g$_{Fe}^{-1}$) and turnover frequency (5.99 e$^{-1}$ s$^{-1}$ site$^{-1}$) obtained from gas diffusion electrode (GDE), which far outperformed those of single pore-sized Fe SACs. In combination with density functional theory (DFT) and ad initio molecular dynamics (AIMD), an inter-site structural heterogeneity-induced effect was revealed, where Fe-N$_4$ sites at mesoporous edges behave as real active sites as a result of electronic modulation of adjacent microporous Fe sites. In situ attenuated total reflectance surface-enhanced infrared absorption spectroscopy (ATR-SEIRAS), in situ Raman and operando XAS measurements provided a thorough understanding of dynamic ORR mechanism on active porous Fe-N$_4$ sites, which underwent a dynamic evolution by breaking Fe-N bond in alkaline medium and therefore lowering reaction barriers. The strong interactions between adjacent pairs of structural heterogeneity Fe sites also contributed to the stabilization of Fe atoms, avoiding the aggregation and migration of active Fe atoms. These findings legitimate rational optimization of SACs by pore structure engineering and provide an in-depth understanding of dynamic ORR mechanism on porous single Fe sites, which might promote the use of SACs in practical applications.

## Results

### Identifying Fe SACs with different pore structure

Highly dispersed Fe SAs-HP catalysts with hierarchical pores were prepared through an encapsulation-pyrolysis-evaporation strategy under NH$_3$ atmosphere as shown in Fig. 1. Here, heme chloride was

allowed to permeate into CTP matrix under a temperature higher than softening point ($T > 120\,°C$) due to strong π-π conjugation between heme chloride and PAHs[19]. Under higher temperature, the soft template, ZnO ($d = 30$ nm), would react with carbon substrate and evaporate, forming abundant mesoporous edge sites to immobilize Fe atoms[20–22]. During this process, NH$_3$ would also react with non-graphitic carbon to readily convert into hydrocarbon species, forming micropores in carbon substrates. Moreover, NH$_3$ could incorporate N atoms into carbon substrate and trap Fe atoms in the microporous sites[23,24]. Herein, highly dispersed Fe SAs-HP catalysts with Fe atoms resided at microcpores and mesopores were achieved without further tedious acid washing process. For comparison, mesoporous-rich Fe SACs (Fe SAs-MSP) was also prepared with the existence of ZnO under N$_2$ atmosphere. Micropore-dominated Fe SACs (Fe SAs-MCP) were synthesized under NH$_3$ atmosphere without ZnO templates. Particularly, defect-free Fe SACs were also prepared without ZnO template under N$_2$ atmosphere (Fe SAs-in plane). Pore structure and Brunauer–Emmett–Teller (BET) specific surface area ($S_{BET}$) of catalysts were then carefully revealed by N$_2$ adsorption-desorption isotherms (Supplementary Fig. 1 and Table 1). The large adsorption capacity of Fe SAs-HP and Fe SAs-MCP at low relative pressure suggested the existence of plentiful micropores, which would facilitate to host Fe atoms. Different from Fe SAs-MCP, Fe SAs-HP, and Fe SAs-MSP exhibited a typical type-IV isotherms with H4 hysteresis loops at high relative pressure ($P/P_0 = 0.4–1.0$), indicating the co-existence of mesopores in Fe SAs-HP. Specific surface area ($S_{BET}$) of Fe SAs-HP increased from 312.7 m$^2$ g$^{-1}$ (Fe SAs-MCP) to 578.9 m$^2$ g$^{-1}$, due to the porogenesis of ZnO sacrificial templates. Pore size distribution demonstrated that Fe SAs-HP exhibited a hierarchical pore structure with micropore and mesopore size distribution centered at 0.47 and 30 nm, respectively. The proportion of micropore area S$_{mcp}$ (S$_{mcp}$/S$_{BET}$) was used as an indicator to quantify the effect of pore size as shown in Supplementary Table 1. Plentiful micropores and mesopores were conducive to immobilizing Fe atoms, strengthening the interactions of adjacent active centers[25]. Thus, the unique Fe SAs-HP with micropores and mesopores densely permeating in the carbon networks was constructed, which could expose active sites and guarantee decent mass and electron transfer at interface.

The Fe contents of as-prepared Fe SACs were 0.72%, 0.43%, 0.20%, and 0.04% for Fe SAs-HP, Fe SAs-MCP, Fe SAs-MSP, and Fe SAs-in plane, respectively, as determined by inductively coupled plasma mass spectrometry (ICP-MS, Supplementary Table 2). Negligible Fe atoms were planted on defect-free nitrogen-doped carbon planes (Fe SAs-in plane) due to insufficient anchoring sites to immobilize Fe atoms. On the contrary, Fe atoms were found to preferentially anchor at micropores and mesoporous edges according to higher Fe contents of Fe SAs-MCP (0.43%) and Fe SAs-MSP (0.20%)[20,26]. A hybrid of Fe moieties stemming from structural heterogeneity was therefore speculated to exist in hierarchically porous Fe SAs-HP, namely defective Fe sites confined in micropores and edge-type Fe sites anchored at mesoporous edges[20]. In addition, Fe contents of Fe SAs-HP (0.72%) were higher than the sum of Fe SAs-MCP and Fe SAs-MSP, suggesting the co-existence of Fe sites at micropores and mesoporous edge would mutually promote the immobilization of Fe atoms, achieving a stable Fe moiety. The highly porous structure of Fe SAs-HP with rich micropores and mesopores was also verified by their largest $I_D/I_G$ values (1.05) detected by Raman spectroscopy (Supplementary Fig. 2).

X-ray diffraction patterns (XRD) of all the catalysts showed only two broad peaks at 25° and 44°, corresponding to (002) and (101) crystal faces of graphitized carbon (Supplementary Fig. 3). No metallic phase can be observed in Fe SACs, indicating the efficient geometry to anchor Fe atoms. The morphology and microstructure of pyrolyzed catalysts were observed by scanning electron microscopy (SEM) and transmission electron microscopy (TEM). The TEM images in Fig. 2a showed that the as-prepared Fe SAs-HP presents a two-dimensional

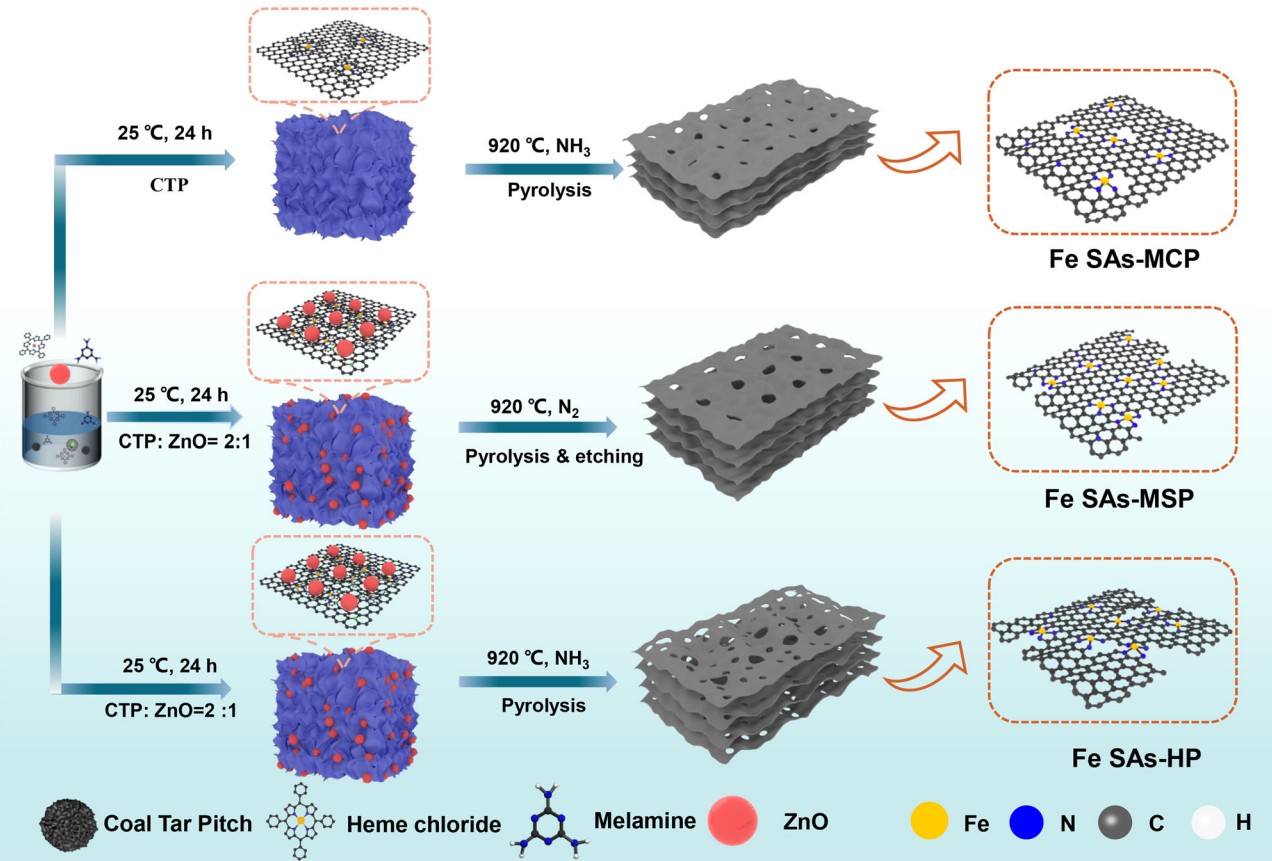

**Fig. 1 | Schematic illustration of catalyst preparation.** Preparation procedures for Fe SACs dominated by micropores (Fe SAs-MCP), mesopores (Fe SAs-MSP), and hierarchical pores of micropores and mesopores (Fe SAs-HP), respectively.

(2D) sheet-like morphology with a wrinkled surface on the edge, while the transparent spots indicate porous feature. The 2D ultrathin nanosheets were also observed in SEM images (Supplementary Fig. 4). The magnified TEM images of Fig. 2b confirmed the porous nature of Fe SAs-HP with a mesopore diameter of ~30 nm, which could be attributed to the porogenesis of ZnO, providing abundant edge sites for immobilizing Fe atoms. The inset image of selected area electron diffraction (SAED) in Fig. 2c exhibited a ring-like pattern, indicating poor crystallinity of Fe SAs-HP. High-resolution transmission electron microscopy (HRTEM) clearly showed the graphene sheets with several few layers. The lattice fringe of Fe SAs-HP was estimated to be 0.349 nm (Fig. 2d, e), which was in line with XRD results $(d_{(002)} = 0.350\ nm)$ and higher than that of graphite (0.335 nm) due to incorporation of Fe atoms. No iron particles could be observed at the whole region of randomly selected TEM images, indicating the high dispersion of Fe moieties.

Atomic force microscopy (AFM, Fig. 2f) linear scan demonstrates that the thickness of these nanosheets is about 4 nm, further revealing the ultrathin graphene-like structure of Fe SAs-HP. High-angle annular dark-field scanning transmission electron microscopy (HAADF-STEM) and corresponding energy dispersive spectrometer (EDS) mapping of Fe SAs-HP (Fig. 2g, h) indicated the Fe moiety was homogeneously dispersed into nitrogen-carbon substrates without agglomeration. To further confirm the atomically dispersed Fe single atoms, an aberration-corrected high-angle annular dark-field scanning transmission electron microscope (AC-HAADF-STEM) was conducted. The Fe atoms were permeated into the carbon matrix without aggregation, indicating their isolated dispersion nature. Along with the mesoporous edges in Fig. 2i, obvious Fe pairs at mesoporous edge and microporous sites can be observed (yellow circles). The magnified AC-STEM images in Fig. 2j at mesopores clearly showed the existence of Fe pairs

originating from structural heterogeneity. The electronic structure of individual metal atoms was found to be redistributed as a result of proximity effect of neighboring metal atoms, which was closely related to adsorption behavior of oxygenated intermediates[25,27]. Thus, the unique Fe pairs with Fe sites resided at adjacent micropores and mesopores in hierarchically porous Fe SACs might modulate electronic structure of active centers through long-range electron regulation and optimized catalytic behavior. The highly disordered structure of Fe SAs-HP was also witnessed by electron paramagnetic resonance (EPR) test, where Fe SAs-HP possessed the highest intensity as shown in Fig. 2k. The same $g$ value of as-prepared porous catalysts corroborated the similar defect types existed in carbon matrix arising from the analogous synthetic conditions[28].

High-resolution X-ray photoelectron spectroscopy (XPS) was conducted to identify the elemental composition and chemical state of as-prepared catalysts. The high-resolution N 1 $s$ spectra in Fig. 3a can be deconvoluted into peaks at 398.0, 399.3, 400.3, 401.0, and 403.6 eV, which can be assigned to pyridinic N, Fe-N, pyrrolic N, graphitic N and oxidized N, respectively. Among various N species, pyridinic N was recognized as responsible for generating isolated Fe-N$_x$ sites, and the graphitic N was conducive to electron transfer in the carbon skeleton[29]. The large contents of edge-type N as shown in Fig. 3b could be attributed to the NH$_3$ and ZnO etching effects, which could facilitate anchoring Fe atoms at micropores and mesoporous edges. Note that the N types in all Fe SAs samples were barely changed as exhibited in Fig. 3b, ruling out its contribution to the intrinsic ORR activity of Fe sites[30,31]. X-ray absorption spectroscopy (XAS) was carried out to decipher the electronic structure and chemical environment of Fe SAs-HP. A pre-edge peak of Fe K-edge X-ray absorption near edge structure (XANES) appeared at 7114 eV, stemming from 1$s$ to 4$p_z$ transition, along with the charge transfer from legend to metal, which

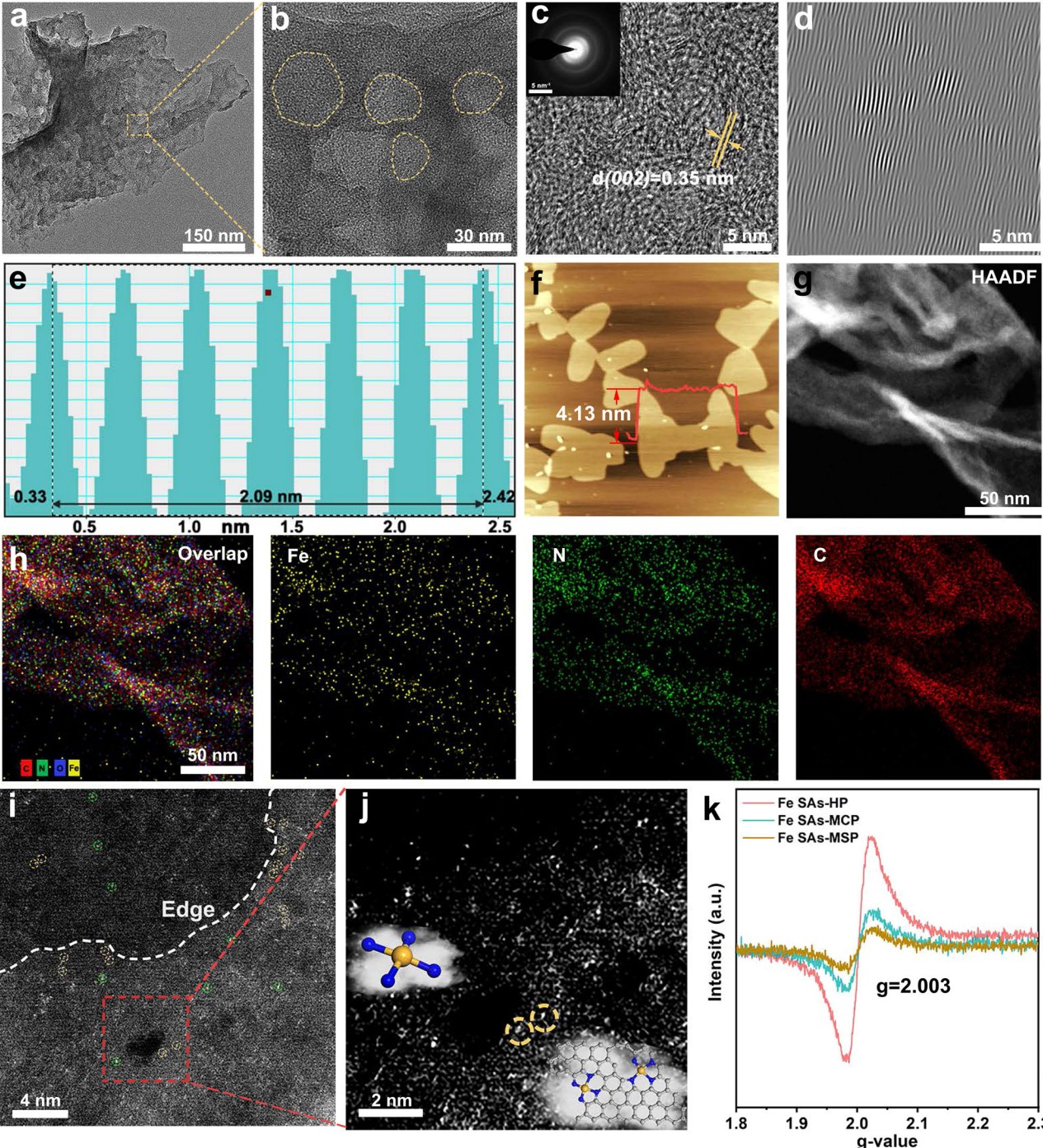

**Fig. 2 | Characterizations of Fe SAs-HP. a** TEM and **b** magnified TEM images of Fe SAs-HP. **c** High-resolution TEM of Fe SAs-HP. The inset picture is the SAED image. **d, e** the analyzed lattice fringes of Fe SAs-HP. **f** AFM images of Fe SAs-HP. The inset curves show the thickness of Fe SAs-HP nanosheets. **g** HAADF-TEM images of Fe SAs-HP. **h** Elemental overlaps and corresponding EDS elemental mapping of Fe, N, and C. **i** AC-STEM of Fe SAs-HP. The white dashed line showed the edge of mesopores. The yellow circles showed Fe atom pairs at micropores and mesoporous edges, and green circles showed the conventional Fe sites inside the carbon matrix. **j** Magnified AC-STEM of Fe SAs-HP. The inset picture shows the atomic structure of Fe sites for Fe SAs-HP. **k** EPR of as-prepared catalysts.

was considered as the fingerprint of porphyrin-like planar Fe-N$_4$. Compared with typical porphyrin-like FePc, an attenuated absorption intensity of pre-edge peak was observed for Fe SAs-HP, suggesting Fe moiety in Fe SAs-HP formed a similar Fe-N$_4$ structure with distorted $D_{4h}$ symmetry[32]. The distorted symmetry could be ascribed to the dominant distorted Fe-N$_x$ sites anchored at micropores and mesoporous edges in Fe SAs-HP. Meanwhile, the energy absorption threshold of Fe SAs-HP was witnessed higher than that of FePc due to a shift to high energy and much closer to that of Fe$_2$O$_3$ as shown in Fig. 3c, implying

the valence state of Fe species in Fe SAs-HP was ~+3, rather than +2. The linear combination method was performed by plotting with the edge absorption energy and valance state of standard samples to obtain further valance state information of Fe[33]. The exact Fe valance state of Fe SAs-HP was calculated to be +2.7, as shown in inset figure in Fig. 3c, which was in line with XPS results (Supplementary Fig. 6). The higher Fe valence state of Fe SAs-HP than typical FePc might be ascribed to the Fe site proximity effect. Simulated differential charge densities were therefore performed to better understand Fe site proximity effect in

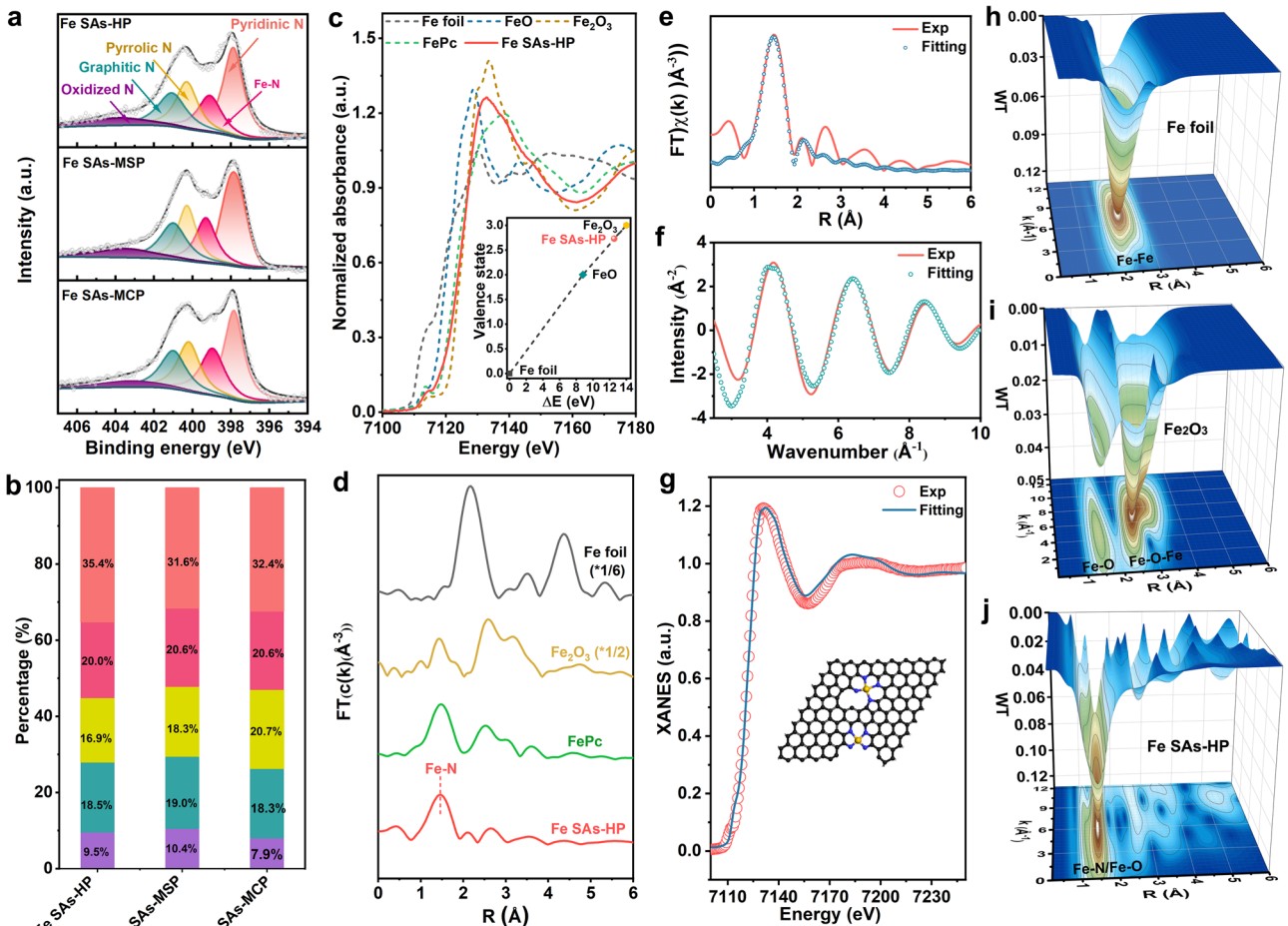

**Fig. 3 | XPS and XAS characterizations of Fe SAs-HP and as-prepared catalysts.**
**a** XPS of N 1 *s* for catalysts and **b** corresponding proportions of N species. Red: pyridinic N. Dark red: Fe-N. Yellow: pyrrolic N. Cyan: graphitic N. Purple: oxidized N. **c** XANES of Fe K-edge and **d** FT-EXAFS of as-prepared catalysts. **e**, **f** Experimental

(Exp) and fitting of Fe SAs-HP at R space and q space, respectively. **g** XANES fitting of Fe SAs-HP. The inset picture indicates the atomic structure of Fe SAs-HP. **h**–**j** WT-EXAFS of Fe K-edge for Fe foil, Fe₂O₃, and Fe SAs-HP, respectively.

hierarchically porous Fe SAs-HP. Mesoporous edge Fe-N₄ sites were found to donate their electron to adjacent microporous Fe-N₄ sites, leading to a higher valence state (Supplementary Fig. 7)[34]. A higher valence state of central Fe atoms was conducive to activate O₂ and strengthen the electron transfer between Fe 3$d$ and O 2$p$ obits[6]. The Fourier transform k³-weighted Fe k-edge extended X-ray absorption fine structure (EXAFS, Fig. 3d) of Fe SAs-HP exhibited a main peak at 1.49 Å, similar to that of FePc, which can be attributed to the first shell of Fe-N scattering path. No peak of Fe-Fe path at 2.2 Å further verified the atomic dispersed Fe moieties in Fe SAs-HP. Quantitative EXAFS fitting of R space analysis indicates the coordination number of Fe moiety was around 4, and the most reasonable coordination structure was Fe-N₄ active sites (Fig. 3e). Further fitting of XANES of Fig. 3g results showed the asymmetric Fe-N₄ structure with Fe atoms lying at micropores defects and edges, which was consistent with Fe K-edge XANES experimental results. Wavelet transform (WT) of Fe k-edge EXAFS oscillations was conducted to further confirm the isolated dispersion of Fe moiety. In accordance with FT-EXAFS, the WT contour plot of Fe SAs-HP shows a maximum intensity at around 5.5 Å⁻¹, corresponding to Fe-N scattering (Fig. 3j). By contrast, Fe foil and Fe₂O₃ standard samples (Fig. 3h, i) exhibited a higher intensity at around 8.0 Å⁻¹, which can be attributed to metallic Fe-Fe scattering. The XAS analysis confirmed the isolated dispersion of Fe-N₄ active sites with asymmetric coordination structure due to incompletely symmetric micropore and mesopore anchoring sites, which were commonly considered active moieties for oxygen electrocatalysis[26,35].

## Electrochemical performance evaluated on RDE

The rotating disk electrode (RDE) measurements were first carried out to estimate ORR activity of as-prepared catalysts using a typical three-electrode system in O₂-saturated 0.1 M KOH. All estimated potentials were referenced to reversible hydrogen electrode (RHE) unless otherwise specified. Impressively, Fe SAs-HP displayed a remarkable ORR activity with onset potential of 1.06 V and a half-wave potential ($E_{1/2}$) of 0.94 V, as shown in Fig. 4a, which far exceeded those of single pore-sized Fe SAs-MCP (0.86 V) and Fe SAs-MSP (0.84 V), illustrating the significant role of hierarchical pore structure. The $E_{1/2}$ of Fe SAs-HP was even 60 mV more positive than benchmark 20% wt Pt/C catalysts. Additionally, Fe SAs-HP also possessed the highest kinetic current density of 28.2 mA cm⁻² at 0.90 V as exhibited in Fig. 4b, which was 21.7 times higher than Fe SAs-MCP (2.8 mA cm⁻²) and 94 times higher than Fe SAs-MSP (0.3 mA cm⁻²), evincing its high catalytic activity and advance in hierarchical pore engineering. The excellent ORR kinetics of Fe SAs-HP were also revealed by its lower Tafel slopes of 84.0 mV dec⁻¹ as shown in Supplementary Fig. 8, which was lower than those of counterparts (85.1, 95.7, 94.0 mV dec⁻¹ for Fe SAs-MCP, Fe SAs-MSP and Pt/C, respectively)[6,36]. The CV curves of Fe SAs-HP and counterparts with scanning rates from 5 to 30 mV s⁻¹ were exhibited in Supplementary Fig. 9. Surprisingly, Fe SAs-MCP possessed the highest electrochemical double-layer capacitance $C_{dl}$ of 26.8 mF cm⁻², and followed by Fe SAs-HP and Fe SAs-MSP. The results suggested that the electrochemical active surface area (ECSA) was not the dominant role for the enhanced ORR activity. The atomically dispersed Fe active sites

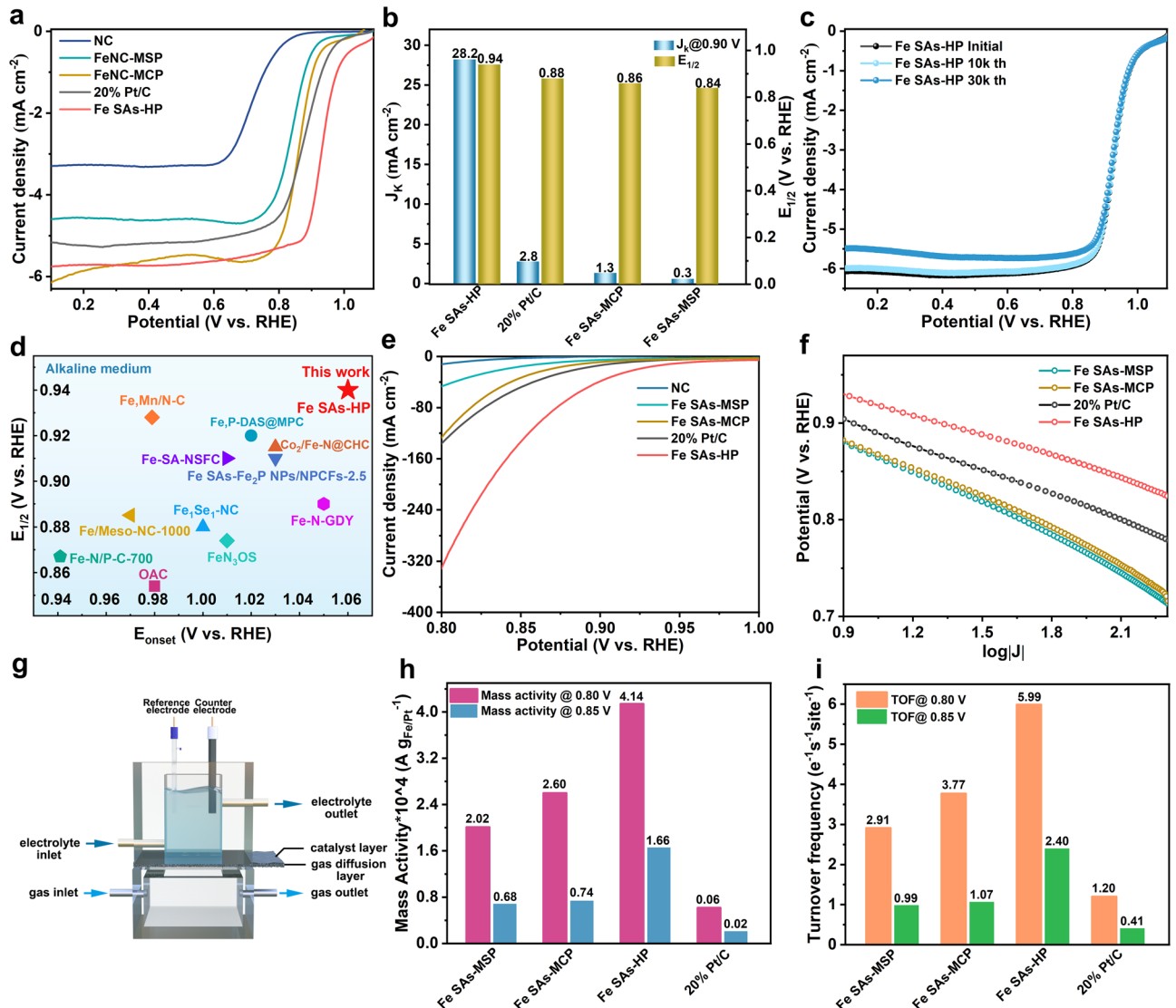

**Fig. 4 | Electrochemical performance of as-prepared catalysts. a** LSV curves of as-prepared catalysts under 0.1 M KOH. **b** Alkaline $J_k$ at 0.90 V and $E_{1/2}$ of catalysts. **c** Alkaline ADT tests of Fe SAs-HP. **d** Comparison of alkaline ORR performance for Fe SAs-HP with reported catalysts, respectively. **e** Polarization curves of as-prepared catalysts with different pore structures by GDE. **f** The plots of potentials against current densities at a logarithmic scale of as-prepared catalysts denoted in **e**. **g** Schematic illustration of GDE measurements. Highly pure oxygen was purged into the chamber with 100% humidity at 70 °C with a flow rate of 150 mL/min. The electrolyte was 1.0 M KOH. **h, i** Mass activity and TOF of as-prepared catalysts at 0.80 V and 0.85 V, respectively.

were also verified by KSCN tests due to SCN⁻ affinity to isolated Fe sites. Fe SAs-HP showed an obvious loss of limiting current density and half-wave potential, indicating atomic Fe sites were the active centers (Supplementary Fig. 10). Inspired by the superior alkaline ORR activity of Fe SAs-HP, acidic ORR activity of as-prepared catalysts was also investigated in 0.1 M HClO₄ as exhibited in Supplementary Figs. 11–13. Fe SAs-HP exhibited acidic $E_{1/2}$ of 0.78 V, which was comparable to that of Pt/C ($E_{1/2}$ of 0.80 V) and was much better than Fe SAs-MCP ($E_{1/2}$ of 0.68 V) and Fe SAs-MSP ($E_{1/2}$ of 0.60 V).

Apart from the ORR activity, stability and selectivity of catalysts are also key indicators to access catalytic performance. The accelerated durability test (ADT) and long-term i-t chronoamperometric tests were performed in both 0.1 M KOH and 0.1 M HClO₄ to evaluate stability of prepared catalysts. Fe SAs-HP exhibited negligible $E_{1/2}$ decay and part loss of limiting current density after 30 k ADT cycles in both alkaline and acidic solution (Fig. 4c and Supplementary Fig. 13), suggesting their exceptional durability and acid resistance ability. The excellent durability was further verified by their i-t chronoamperometric tests (Supplementary Fig. 14). Fe SAs-HP displayed 97.3%

current retention after 50,000 s tests in 0.1 M KOH, superior to those of Fe SAs-MCP (73.6 %), Fe SAs-MSP (69.8%) and Pt/C (58.9%). The excellent durability of active sites in Fe SAs-HP might benefit from the strong electron interactions between adjacent mesoporous and microporous Fe sites, as verified by differential charge densities and AC-STEM images. No obvious changes in morphology of Fe SAs-HP were monitored after 30 k ADT according to TEM characterizations (Supplementary Fig. 15), validating the advantages of unique hierarchically porous structure. The same scenarios were also witnessed for Fe SAs-HP during harsh i-t tests at high overpotentials of 0.2 V. However, numerous tiny clusters and noticeable leaching of Fe atoms were observed for Fe SAs-MCP and Fe SAs-MSP, as shown in Supplementary Figs. 16, 17 and Table 3, respectively. The strong interactions between adjacent pairs of structural heterogeneity Fe sites was conducive to stabilizing Fe atoms and avoided the aggregation and migration of active Fe atoms. The electron transfer number ($n$) of Fe SAs-HP was calculated based on Koutecky–Levich (K-L) equation. The n values of Fe SAs-HP were determined to be 4.02-4.21 at potential range from 0.65 to 0.85 V as shown in Supplementary Fig. 18. Selectivity and

transferred electron number measurements were further estimated on a rotating ring disk electrode (RRDE, Supplementary Fig. 20) in 0.1 M KOH. Fe SAs-HP exhibited a low $H_2O_2$ yield (<5%) at long-range potentials from 0.2 V to 0.8 V and high selectivity of 4-e$^-$ ORR of (3.96–4.00), indicating a favorable 4e$^-$ ORR pathway on unique Fe-N$_4$ active sites. The methanol crossover effect was determined by instantaneously injecting methanol into $O_2$-saturated 0.1 M KOH solution during i-t chronoamperometric tests. When injecting methanol at 500 s, Fe SAs-HP exhibited no disturbance of current, as demonstrated in Supplementary Fig. 21, while Pt/C suffered from a sharp current loss, demonstrating the excellent methanol tolerance ability of active Fe sites in Fe SAs-HP. The superior ORR activity of Fe SAs-HP also surpassed most reported high-activity Fe-based catalysts, as shown in Fig. 4d and Supplementary Fig. 22 and Tables 4 and 5, where the ORR activity of Fe SAs-HP lies at the upper right corner of activity maps in both alkaline and acidic solutions. Compared to single pore-sized Fe SAs-MCP and Fe SAs-MSP catalysts as shown in comprehensive Supplementary Fig. 23, Fe SAs-HP exhibited enhanced ORR activity, durability, and selectivity, which might be ascribed to interactions of hierarchical active Fe sites.

Encouraged by the remarkable ORR catalytic activity, a home-made liquid Zn-air battery (ZAB) was prepared to verify the promising applications of Fe SAs-HP catalyst in energy storage configurations with Fe SAs-HP as air cathodes, zinc as anodes, and 6.0 M KOH + 0.2 M Zn(Ac)$_2$ as electrolyte. For comparison, Zn-air batteries with 20% Pt/C catalysts as cathodes were also assembled. Zn-air battery delivered a maximum peak power density of 254 mW cm$^{-2}$ at a current density of 398 mA cm$^{-2}$ as exhibited in Supplementary Fig. 24, which outperformed the benchmark Pt/C catalyst (164 mW cm$^{-2}$). The peak power density of ZAB assembled with Fe SAs-HP was also prominent among recently reported liquid ZABs as exhibited in Supplementary Table 4. $H_2$-$O_2$ proton exchange membrane fuel cell was also fabricated to investigate the performance of as-prepared catalysts as exhibited in Supplementary Fig. 25. $H_2$-$O_2$ fuel cell with Fe SAs-HP as air cathodes achieved a maximum peak power density of 449 mW cm$^{-2}$ at current density of 1360 mA cm$^{-2}$, which was 3.2 times higher and 7.6 times higher than those of Fe SAs-MCP (142 mW cm$^{-2}$) and Fe SAs-MSP (59 mW cm$^{-2}$), indicating the efficient activity of Fe sites originating from hierarchically porous structure.

## Evaluation of intrinsic site activity

The mass transport in RDE measurements is severely limited by oxygen solubility in the electrolyte, resulting in a mass transport-controlled performance. GDEs enable the reactants transport similar to the active sites[37]. Hence, a home-made GDEs were employed to evaluate the intrinsic MA and turnover frequency (TOF) of Fe active sites[25,38]. Polarization curves of all samples were recorded on GDE under tight controls of gas flow, temperature and humidity as illustrated in Fig. 4g. Figure 4e of voltammograms exhibited typical kinetically controlled reduction process. The recorded current density of Fe SAs-HP increased greatly with applied overpotentials. In particular, NC catalysts exhibited negligible reduced current on GDE tests, suggesting that the ORR activity mainly arose from Fe moieties. Besides, plots of potentials against current densities in logarithmic coordinates (log J) as shown in Fig. 4f for Fe SACs exhibited a linear relationship at potentials from 0.80 V to 0.90 V, indicating the oxygen mass transport is sufficient enough to achieve pure ORR kinetic current provided by Fe active sites.

MA and turnover frequency at 0.80 V and 0.85 V were selected to access the intrinsic Fe site activity[39,40]. MA of as-prepared Fe SACs was obtained from apparent current normalized by total weight of Fe contents loaded on GDE based on ICP-MS results. Fe SAs-HP delivered the MA of $4.14 \times 10^4$ A g$_{Fe}^{-1}$ at potential of 0.80 V as shown in Fig. 4h, which was 1.6 and 2.0 times higher than single pore-sized Fe SAs-MCP and Fe SAs-MSP and remarkable among recently reported non-precious catalysts as shown in Supplementary Table 6[25,40]. The results imply that the

intrinsic Fe site activity relies heavily on the geometry of Fe-N$_4$. A hybrid Fe sites in Fe SAs-HP with Fe sites anchored on micropores and mesoporous edges were witnessed to be more efficient than single Fe sites when ruling out the mass transport factor in GDE tests, which might be due to inter-site induced optimization of microporous and mesoporous Fe sites.

Turnover frequency was then calculated to quantitative understand how heterogeneity Fe sites affect intrinsic activity. The TOF characterized the unit electrons transferred per active site and per second[39,40]. Based on the kinetic current normalized by the number of Fe active sites, the TOF demonstrates a tight relationship with the geometry of Fe sites. Specifically, individual Fe sites for Fe SAs-HP achieved the highest activity with TOF of 5.99 e$^{-1}$ s$^{-1}$ site$^{-1}$ at potential of 0.80 V as exhibited in Fig. 4i, which was 1.6 times and 2.1 times higher than the Fe sites of Fe SAs-MCP and Fe SAs-MSP. The exceptional site activity of Fe SAs-HP also surpassed most reported SACs as exhibited in Supplementary Table 6[25,40,41]. The TOF for Fe SAs-HP was also excellent compared to Fe SAs-MCP and Fe SAs-MSP at potential of 0.85 V, which was 2.2 times and 2.4 times higher than those of counterparts. The TOF results strongly indicated individual Fe sites of Fe SAs-HP were more active than those of single pore-sized Fe SAs-MCP and Fe SAs-MSP, which could be largely probably ascribed to the strong interactions of Fe sites induced by structural heterogeneity in hierarchically porous Fe SAs-HP matrix.

## Theoretical evidence of structural heterogeneity induction effect

DFT calculations were conducted to understand the structural heterogeneity-induced optimization of individual Fe sites in hierarchically porous Fe SAs-HP. Several types of Fe-N$_4$ sites including Fe-N$_4$ in the plane of carbon matrix (FeN$_4$-in plane), edge-type Fe-N$_4$ at mesoporous edges (Fe SAs-HP) and Fe-N$_4$ at microporous and mesoporous edges (Fe SAs-HP), were selected as prototypes to understand ORR catalytic process as shown in Supplementary Figs. 26–30. Considering the relative high contents of microporous Fe sites in Fe SAs-MCP, the inter-site interaction of individual microporous Fe sites was also performed as exhibited in Supplementary Fig. 28. Catalytic behavior of Fe sites displayed a remarkable correlation to their geometry as shown in Fig. 5b. Activation of $O_2$ were found to be unfavorable on microporous Fe sites of Fe SAs-MCP due to uphill free energy of 0.73 eV at first ORR steps. And the desorption of *OH was regarded as rate determine step (RDS) for edge-type mesoporous Fe sites, in plane Fe-N$_4$ and hybrid Fe sites at edge of mesopores (Fe SAs-HP@MSP). At $U = 0.46$ V, the free energy difference of Fe SAs-HP@MSP site in the RDS is 0.68 eV, which was lower than those of FeN$_4$-in plane (1.17 eV) and single mesoporous Fe sites in Fe SAs-MSP (0.85 eV). In comparison, Fe-N$_4$ at microporous sites for Fe SAs-HP (Fe SAs-HP@MCP) was also calculated. The first step of $O_2$ activation on microporous sites (Fe SAs-HP@MCP) was determined as RDS and showed a huge free energy difference of 2.05 eV, which was unfavorable to initiate ORR. The Fe-N$_4$ sites at mesoporous edges adjacent microporous Fe-N$_4$ were therefore regarded as active sites in Fe SAs-HP. The optimized structure of dual heterogeneity Fe sites was exhibited in Fig. 5a, where a Fe-N bond belonging to mesoporous Fe-N$_4$ active site will break to Fe-N$_3$ coordination structure due to strong interactions between microporous Fe-N$_4$ and mesoporous Fe-N$_4$. However, the optimized structure of dual site system for Fe SAs-MCP in Supplementary Fig. 28 remained barely changed. Further theoretical results of differential charge densities in Fig. 5c and Supplementary Fig. 31 disclosed that the neighboring microporous Fe sites would regulate adsorption behavior of intermediates and reduce reaction barriers of RDS for mesoporous Fe sites, thereby accelerating ORR kinetic process. After adsorption of *OH intermediates, central Fe atoms at single mesoporous edges would lose 1.38 e$^-$ and the adsorbed *OH groups will obtain 0.50 e$^-$. Different from scenario of single

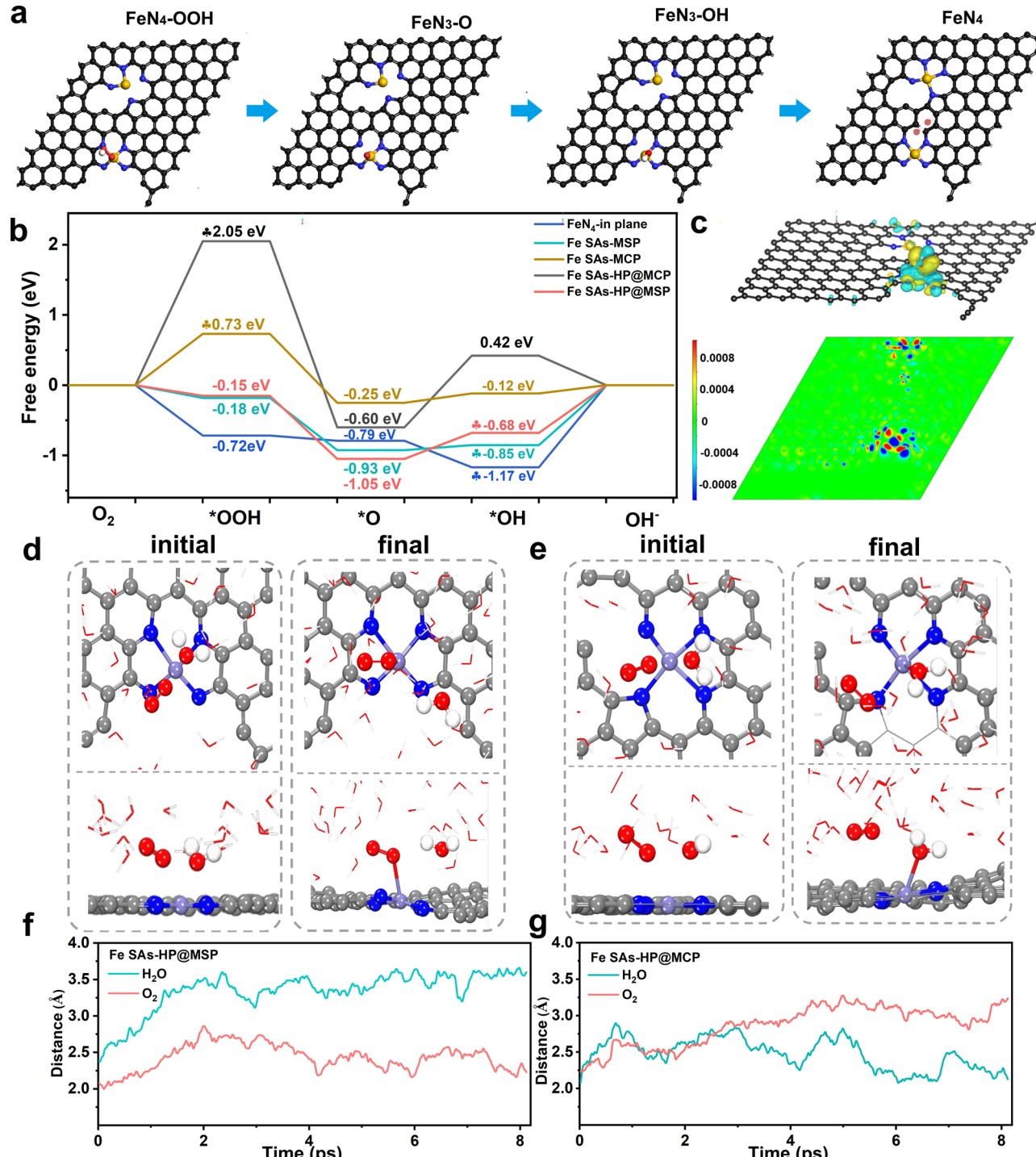

**Fig. 5 | Theoretical calculations of as-prepared catalysts. a** Optimized structure of Fe SAs-HP during ORR process. **b** Free energy against the reaction pathway of as-prepared catalysts. **c** Charge distribution of Fe SAs-HP after OH adsorbed. **d, e** Initial and final configuration of the 8 ps AIMD simulations for mesoporous Fe sites (Fe SAs-HP@MSP) and microporous Fe sites (Fe SAs-HP@MCP), respectively. **f, g** the distance between Fe sites and O atom in adsorbed $H_2O$ (green) and $O_2$ (red) as a function of time of AIMD simulations at 298 K.

mesoporous Fe-$N_4$, introduction of microporous Fe-$N_4$ adjacent to mesoporous $FeN_4$ would regulate the adsorption behavior of *OH intermediates. The mesoporous Fe atoms adjacent to microporous Fe-$N_4$ sites will lose 1.36 e$^-$ and *OH obtained 0.39 e$^-$. The much more stable adsorption of *OH on single mesoporous Fe-$N_4$ sites will lead to unfavorable desorption steps of *OH intermediates. On the contrary, the incorporation of adjacent microporous Fe-$N_4$ sites would optimize adsorption of *OH on active centers and facilitate formation of OH$^-$,

which was consistent with free energy plots. Therefore, Fe-$N_4$ sites at mesopores in the hybrid sites were recognized as active sites as a result of electronic modulation by adjacent microporous Fe-$N_4$ sites. Other Fe moieties anchored at microspores or mesoporous edges will behave as conventional active sites. The projected crystal orbital Hamilton population (COHP) was performed to understand interaction between *OH of RDS process and active centers[27]. And integrated COHP (ICOHP) is obtained by integrating the energy from minus infinity to

Femi level, where a linear relationship between ICOHP and $\Delta E_{ads}$*OH was achieved. The linear relationship provided a quantitative understanding of the adsorption of *OH on the central metal atoms. Compared with single mesoporous Fe sites and microporous Fe sites (Supplementary Fig. 32), Fe-N$_4$ anchored at mesoporous edges adjacent to microporous exhibited a proper adsorption of *OH, guaranteeing a balance between activity and desorption of OH⁻. The theoretical results clearly revealed the inter-site induced effect originated from structural heterogeneity, where the microporous Fe-N$_4$ site existed as a modulator to mesoporous Fe sites.

AIMD simulations were conducted to gain an in-depth insight of ORR behavior on Fe SAs-HP. An explicit solvent model was use to assess solid-liquid interactions by introducing OH⁻ and H$_2$O molecules as shown in Supplementary Fig. 33. In addition, since all AIMD simulations are performed under constant charge conditions, the charge extrapolation method developed by Chan and Nørskov is applied for constant potential corrections[42,43]. A production period of 8 ps was used to evaluate the solid-liquid interaction between active sites and solvents. The initial and final configuration of the 8 ps AIMD simulation on mesoporous sites (Fe SAs-HP@ MSP) and adjacent microporous sites (Fe SAs-HP@MCP) were shown in Fig. 5d, e. The mesoporous Fe sites (Fe SAs-HP@ MSP) preferentially adsorbed O$_2$ molecules as exhibited in Fig. 5d, f, indicating a favorable oxygen activation steps and was also consistent with reaction-free energies in Fig. 5b. Different form the mesoporous Fe sites, H$_2$O molecules tended to be adsorbed on microporous Fe sites during the production due to the shortened Fe-O distance between Fe sites and H$_2$O molecules as shown in Fig. 5e, g, demonstrating an inferior ORR dynamic and O$_2$ activation process on microporous Fe sites. The energy and temperature fluctuated within a certain range, suggesting the structure of dual Fe sites system is stable during ORR process as shown in Supplementary Fig. 34, suggesting the structure of dual Fe sites system is stable during ORR process. The Fe-N$_4$ site on mesoporous edge in the dual-site system behaved more active and was deemed as active sites during ORR process.

Based on simulations above, Fe-N$_4$ sites on mesoporous edge were identified as active sites in Fe SAs-HP as a result of inter-site structural heterogeneity induced optimization by adjacent microporous Fe-N$_4$ sites. The adjacent microporous Fe sites would function as a modulator to mesoporous active Fe centers by regulating their electronic structure, which facilitates activation of O$_2$ and desorption of key *OH intermediates of RDS process of and lowering energy barriers. Thus, an inter-site structural heterogeneity-induced optimization of active Fe sites was revealed, which accounted for the structural origin of enhanced intrinsic activity for hierarchically porous Fe-N$_4$ sites in comparison with single pore-sized Fe SACs.

## Investigation of dynamic mechanisms on active Fe-N$_4$ sites

The structure dynamic evolutions will determine the catalytic behavior of active centers. An in-depth understanding of dynamic evolution of porous Fe-N$_4$ sites would in turn guide rational design of efficient Fe SACs. Herein, in combination with in situ ATR-SEIRAS, in situ Raman and operando XAS measurements, structure evolution of hierarchically porous Fe-N$_4$ sites was unclosed. An increasing intensity of absorbance at 3450 cm⁻¹ with applied overpotentials in ATR-SEIRAS measurements can be observed under working conditions in 0.1 M KOH as shown in Fig. 6a, which can be ascribed to the stretching vibration mode of adsorbed OH group. The accumulated hydroxyl groups on active Fe sites indicated a blocked desorption of OH, in line with RDS process of Fe SAs-HP in theoretical calculations[44]. The same occasion was also monitored under acidic working conditions as shown in Supplementary Fig. 35. And an increasing peak at 1650 cm⁻¹ can be assigned to bending vibration mode of hydroxyl in H$_2$O molecules, which might be ascribed to the in situ adsorbed H$_2$O reactants in alkaline medium and the formed H$_2$O product under acidic medium

during ORR process. Considering the limited evidence provided by solo characterization, in situ Raman measurements were also performed to detect structure evolution and key intermediates. Under both alkaline and acidic working conditions, an intense peak located at 1250 cm⁻¹ can be attributed to the adsorption of *OH as shown in Fig. 6b and Supplementary Fig. 36, which was in line with in situ ATR-SEIRAS observations. The intense signals of adsorbed *OH suggested a hampered desorption behavior of *OH in RDS process. An extra peak at 940 cm⁻¹ was observed in acidic Raman spectra in Supplementary Fig. 34, which can be assigned to the adsorption of ClO$_4$⁻[45]. The intensity ratios of D-G band ($I_D/I_G$) for Fe SAs-HP were surveilled to incline to higher values with the increase of overpentials, indicating a dislocation of carbon matrix. And the value of $I_D/I_G$ in alkaline medium was found much higher than those in acidic medium (Fig. 6c and Supplementary Fig. 37), suggesting a sharp structure dislocation under alkaline working conditions, which might be ascribed to the breaking of Fe-N bond. The slightly increased $I_D/I_G$ value in acidic solution might be caused by the adsorbed axial intermediates, which will drive central Fe atoms out of N-4 plane and form five-coordinate Fe-N$_4$(OH)[46].

The ex-situ results revealed a distorted Fe-N$_4$ coordination structure with a valence state of +2.7 as a result of electronic regulation by adjacent microporous Fe sites. However, understanding catalytic behavior of active Fe-N$_4$ sites with porous features under working conditions remained a challenge. Here, operando XAS measurements (Fig. 6d–f) were conducted to give detailed coordination structure evolution and valence transformation of porous Fe active centers. In 0.1 M KOH, the valence state of central Fe increased to +2.83 under open-circuit voltage (OCV), which might be due to the adsorbed oxygenated groups such as OH⁻, O$_2$ or H$_2$O molecules. According to previous reports, a higher valence state of central Fe atom due to pre-adsorbed reactants was conducive to O$_2$ activation on active Fe sites[47,48]. With applied overpotentials in alkaline environment, the Fe-N coordination number (CN) was monitored to decrease from 3.9 to 2.9 (Fig. 6f), indicating a rupture of Fe-N bond during the electrochemical process, in good agreement with optimized structure in theoretical calculations as a result of strong electronic interactions between adjacent mesoporous and microporous Fe-N$_4$ sites. The observations were also well consistent with in situ Raman measurements. At the same time, the CN of Fe-O bond was found to increase from 0 to 0.8, suggesting a strong interaction between Fe and adsorbed oxygenated species. Thus, structure evolution of active Fe-N$_4$ sites was surveilled to undergo a transformation from Fe-N$_4$ coordination structure to Fe-N$_3$O(H). Meanwhile, the valence state of central Fe atoms was decreased from +2.83 to +2.33 due to the break of Fe-N bond. A lower valence state of central atoms arising from dynamic reconstruction of coordination structure in unique hierarchically porous Fe-N$_4$ sites could optimize the adsorption/desorption of oxygenated intermediates during ORR process, which accounted for their favorable ORR reaction pathway and remarkable activity of Fe SAs-HP[14,40]. Different from scenarios under alkaline conditions, the XANES of Fe K-edge remained barely changed under working conditions, suggesting a stable coordination structure of Fe sites as exhibited in Supplementary Figs. 38–40. In combination with optimized catalytic structure of theoretical calculations and in situ/operando measurements, structural evolutions of active Fe-N$_4$ moieties were speculated to undergo a Fe-N$_4$ to Fe-N$_3$ structure switch due to strong interactions of unique mesoporous and microporous pair sites in hierarchically porous Fe SAs-HP, which regulated activation and adsorption behaviors of reactants during ORR process. The schematic illustration of the dynamic behavior for active porous Fe sites during ORR under alkaline solutions was illustrated in Fig. 6g.

For comparison, Operando XAS and in situ Raman of Fe SAs-MCP and Fe SAs-MSP were also performed to monitor the dynamic switching behavior of active Fe sites. Interestingly, the dynamic switching behaviors of microporous and mesoporous Fe sites were

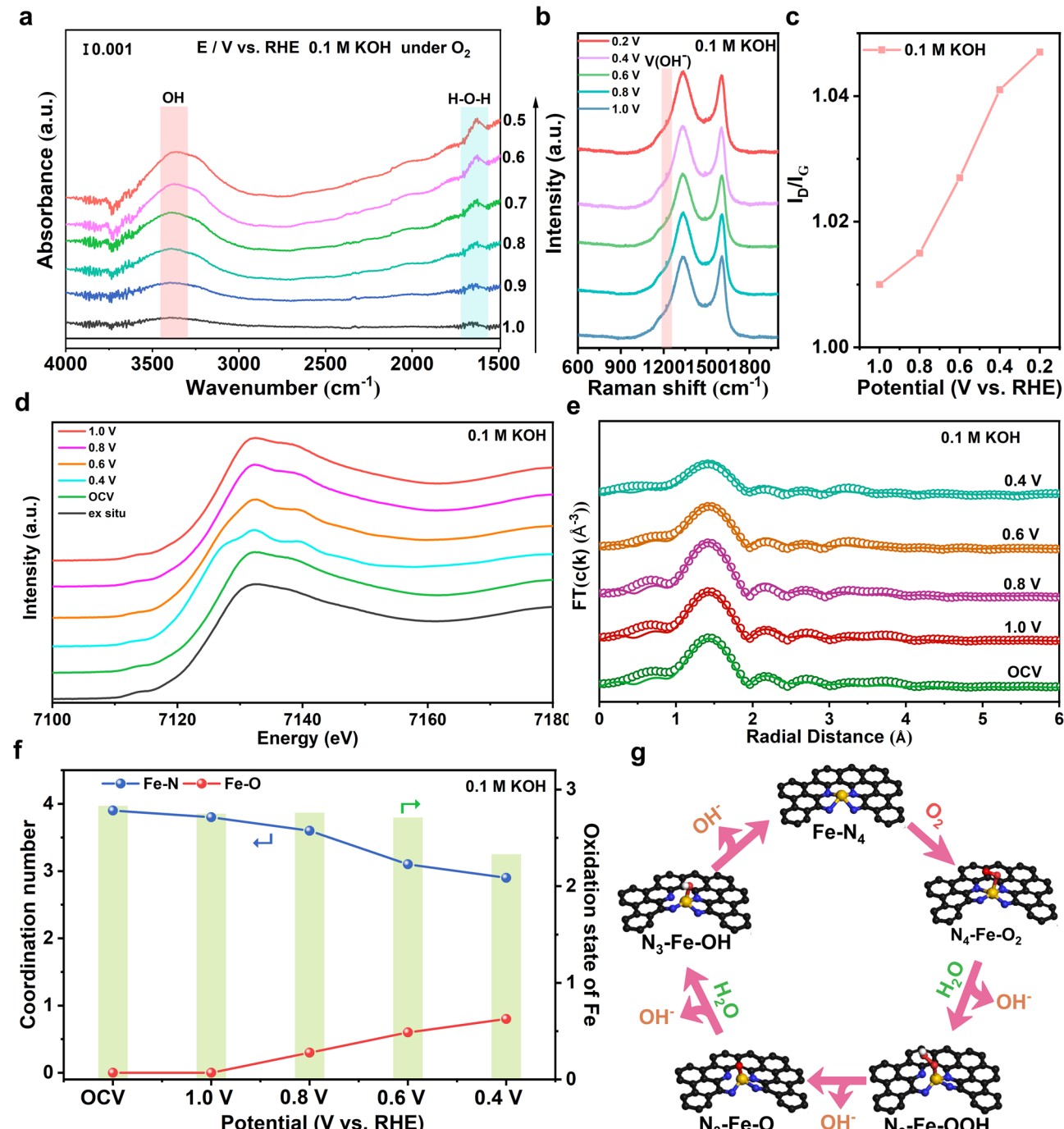

**Fig. 6 | In situ and operando characterizations of Fe SAs-HP in alkaline solutions. a** In situ ATR-SEIRAS of Fe SAs-HP under alkaline media. **b, c** In situ Raman spectra of Fe SAs-HP tested in 0.1 M KOH and corresponding $I_D/I_G$ values with applied overpotentials. **d, e** Operando XANES of Fe K-edge for Fe SAs-HP and corresponding FT-EXAFS fitting analysis under 0.1 M KOH. **f** Corresponding coordination number for Fe-N and Fe-O as well as oxidation state of Fe under alkaline working conditions. **g** Dynamic evolutions of Fe SAs-HP under alkaline working conditions.

quite different from each other. With the increasing of applied overpotentials in alkaline environment as shown in Supplementary Fig. 41 and Table 9, the Fe-N coordination number (CN) of Fe SAs-MCP was monitored to decrease from 3.8 at OCV to 1.8 at higher overpotentials of 0.4 V, indicating an evolution of Fe-N$_4$ to Fe-N$_2$. Meanwhile, the CN of adsorbed intermediates on microporous Fe sites (Fe-O$^1$) was observed to increase from 0 to 0.5 at 0.6 V. Impressively, new Fe-Fe scattering path can be surveilled at ~2.5 Å under 0.4 V as shown in Supplementary Fig. 39b, suggesting the scattering path of FeO. The CN of Fe-Fe and the Fe-O$^2$ (lattice oxygen) was 6.1 and 3.6, respectively.

Note that the standard CN for Fe-O and Fe-Fe was 6 and 12, respectively. The reduced CN for FeO could be ascribed to the formation of tiny cluster[49]. The results indicated that the microporous Fe-N$_4$ sites tended to break Fe-N bond and aggregate into FeO tiny clusters under higher overpotentials, which accounted for their loss of current density during long-term durability tests. In addition, the intensity ratios of D-G band ($I_D/I_G$) for Fe SAs-MCP were observed to incline to higher values with the increase of overpentials, demonstrating a dislocation of carbon matrix as shown in Supplementary Fig. 42. In the case of Fe SAs-MSP, the dynamic structural evolutions were surveilled to

undergo a transformation from Fe-N$_4$ to Fe-N$_2$ with the increasing of applied overpotentials from 1.0 V to 0.4 V as demonstrated in Supplementary Fig. 43. The dramatically structure deformation was also verified by in situ Raman results due to the increased $I_D/I_G$ as shown in Supplementary Fig. 44. The unsaturated Fe atoms were vulnerable to oxygenated intermediates and might leach out from N$_4$ plane, which would cause great loss of current density during durability tests.

## Discussion

In summary, we have constructed hierarchically porous Fe single atoms catalysts with Fe atoms residing at the micropore and mesopore edges and identified the key role of inter-site structural heterogeneity-induced effect in optimizing catalytic performance. Fe SAs-HP exhibited superior ORR activity and durability in both alkaline and acidic media. Compared to single pore-sized Fe SACs, Fe SAs-HP delivered a high MA of $4.14 \times 10^4$ A g$_{Fe}^{-1}$ and site activity due to the strong electron modulation of adjacent microporous Fe-N$_4$. DFT calculations and AIMD simulations revealed the inter-site structural heterogeneity-induced effect, where mesoporous Fe-N$_4$ sites were the active sites as a result of electronic regulation by adjacent microporous Fe sites. The existence of hierarchically microporous Fe-N$_4$ sites would facilitate the desorption of *OH in RDS and lower reaction barriers on adjacent active centers. More importantly, in combination with in situ ATR-SEIRAS, in situ Raman, and operando XAS measurements, the active Fe-N$_4$ sites were surveilled undergoing dynamic structure reconstruction from Fe-N$_4$ to FeN$_3$(OH) by breaking Fe-N bond in alkaline solution, which was conducive to optimizing desorption of intermediates. While in an acidic medium, the active Fe-N$_4$ sites were monitored to stay stable, explaining their excellent long-term durability under acidic solutions. This work sheds new light on the inter-site structural heterogeneity-induced optimization of central metal atoms in hierarchical porous structure at atomic-level precision and provides a thorough understanding of dynamic evolution for porous Fe sites, paving the way for developing efficient catalysts for practical applications.

## Methods

### Materials
All the chemical materials were used as obtained without further purification. Melamine C$_3$H$_6$N$_6$ (AR, 99%), heme chloride C$_{34}$H$_{32}$ClN$_4$O$_4$Fe (AR, 95%), Potassium hydroxide KOH (Electronic grade, 99.999%), N,N-dimethylformamide (DMF, AR, 99.5%), zinc oxide ZnO (30 nm, AR, 99.9%) and silica SiO$_2$ (30 nm, AR, 99.9%) were purchased from Shanghai Aladdin Biochemical Technology Co., Ltd. CTP was obtained from Hebei Feitaiyuan Energy Technology Limited Company.

### Preparation of Fe SAs with micropore structure (Fe SAs-MCP)
Fe SAs-MCP was prepared through a pyrolysis process. In a typical synthesis, 1 g of CTP and 2 g of melamine were dispersed into 25 mL DMF under stirring to form solution A. 25 mg of heme chloride was dispersed into 25 mL DMF under sonication to form solution B. Subsequently, solution B was dropwise into solution A and stirred for 24 h. Thereafter, the mixed solution was heated at 120 °C to vapor dissolvent. Then the obtained precursors were heated at a temperature of 920 °C for 2 h with a heating rate of 2 °C/min under NH$_3$ atmosphere. Before the heating process, N$_2$ was used to exclude the air. The as-prepared catalysts were labeled as Fe SAs-MCP.

### Preparation of Fe SAs with mesopore structure (Fe SAs-MSP)
Similar to the synthesis procedure of Fe SAs-MCP, 0.5 g of ZnO with a diameter of 30 nm was selected as a sacrificial template and added into solution A. The obtained precursors were then heated at a temperature of 920 °C for 2 h with a heating rate of 2 °C/min under N$_2$ atmosphere. The as-prepared catalysts were labeled as Fe SAs-MSP.

### Preparation of Fe SAs with hierarchical pores (Fe SAs-HP)
The preparation process of Fe SAs-HP was similar to Fe SAs-MCP, except that 0.5 g of ZnO with diameter of 30 nm were selected as a soft template and added into solution A. The obtained precursors were then heated at a temperature of 920 °C for 2 h with a heating rate of 2 °C/min under NH$_3$ atmosphere. The as-prepared catalysts were labeled as Fe SAs-HP.

### Electrochemical measurements
All electrochemical measurements were conducted on a pine workstation (pine instrument company) with a standard three-electrode system. The glassy carbon electrode with a diameter of 5 mm and graphite rod were employed as working electrode and counter electrode, respectively. The saturated calomel electrode (SCE) and Ag/AgCl electrode were used as reference electrode under alkaline medium and acidic medium, respectively. The catalyst inks were prepared by adding 5 mg of catalysts into 0.5 mL ethanol, containing 10 μL of 5% wt. Nafion and followed by sonication for 1 h. The electrochemical measurements were conducted at room temperature. Typically, 20 μL of catalyst ink was casted on the electrodes, and the mass loadings were kept at 1.0 mg/cm$^2$. The same standard was also used for commercial Pt/C catalyst. Pure O$_2$ was introduced into electrolytes before ORR tests. The linear sweep voltammetry (LSV) curves of all samples were obtained in O$_2$ saturated 0.1 M KOH or 0.1 M HClO$_4$ at a sweep rate of 5 mV/s with IR$_s$ compensations. The resistance was manually compensated, and the resistance was determined by pine system under impedance spectroscopy mode. The electrolyte was measured by pH instruments to ensure a constant test environment (pH = 13 for alkaline and pH = 1 for acidic medium). All the measured potentials were converted to RHE based on the equations:

$$E(RHE) = E(SCE) + 0.241 + 0.059*pH - IR_s \ (0.1\,\text{M KOH}) \quad (1)$$

$$E(RHE) = E(Ag/AgCl) + 0.197 + 0.059*pH - IR_s \ (0.1\,\text{M HClO}_4) \quad (2)$$

Where $E(RHE)$ is the potential vs reversible hydrogen electrode, $E(SCE)$ and $E(Ag/AgCl)$ are the measured potential with SCE and Ag/AgCl as reference electrodes. pH is the hydrogen ion concentration of electrolyte. $I$ is the measure current, and Rs is the compensated solution resistance.

The electrochemical double-layer capacitance $C_{dl}$ was determined at a sweep rate from 5 mV/s to 40 mV/s for CV tests in non-Faraday range of 1.11 V to 1.01 V. The $C_{dl}$ was calculated according to the equations:

$$C_{dl} = \frac{I_c}{\upsilon} \quad (3)$$

Where $I_c$ is the current density (mA cm$^{-2}$) and $\upsilon$ is the scan rate (mV s$^{-1}$). The kinetic current density ($J_k$) was evaluated according to Koutecky–Levich equation:

$$\frac{1}{J} = \frac{1}{J_L} + \frac{1}{J_k} = \frac{1}{B\omega^{1/2}} + \frac{1}{J_K}$$
$$B = 0.62nFC_0D_0^{2/3}V^{-1/6} \quad (4)$$

Where $J$, $J_L$, and $J_k$ are the measured current density, limiting current density and kinetic current density, respectively. $\omega$ is the angular velocity of rotating disk electrode, $n$ is the transferred electron number, $F$ is Faraday constant (96,485 C mol$^{-1}$), $C_O$ is the bulk concentration of O$_2$ ($1.2 \times 10^{-6}$ mol cm$^{-3}$), $D_O$ ($1.9 \times 10^{-5}$ cm$^2$ s$^{-1}$) is the diffusion coefficient of O$_2$ in 0.1 M KOH solution. And, $V$ is the kinematic viscosity of the electrolyte (0.01 cm$^2$ s$^{-1}$). The electron transferred number at different potentials were obtained by Koutecky-Levich equations under different rotation speeds at 400, 625, 900, 1225, 1600 and 2025 r/min. The rotation speeds were converted into angular velocity ($\omega$).

$H_2O_2\%$ yield and the electron transfer number ($n$) are estimated by Eqs. (5) and (6) according to RRDE tests:

$$H_2O_2\% = \frac{200*I_r}{I_r + N*I_d} \quad (5)$$

$$n = \frac{4*I_d}{I_d + I_r/N} \quad (6)$$

Where $I_r$ and $I_d$ are the ring current and disk current, respectively. $N = 0.37$ is the current collection efficiency of Pt ring.

Accelerated durability tests were performed with a sweep rate of 2000 mV/s from 1.0 V to 0.55 V (vs. RHE). Long-term i-t chronoamperometric tests were carried out on an RDE with a rotate speed of 200 r/min at a constant voltammetry of 0.75 V and 0.55 V (vs. RHE) for 50,000 s under alkaline and acidic medium, respectively. A harsh stability test was performed by loading ink of as-prepared catalysts on carbon paper (1 cm$^{-2}$, Sigracet carbon paper, 28BC) at 0.2 V vs. RHE with high overpotentials for 8 h in 0.1 M KOH under a standard three electrodes system. The Fe loadings and the volume of electrolytes were kept same. Fe loadings were kept at 0.0145 mg cm$^{-2}$ based on ICP-MS result, and the volume of electrolyte was kept for 20 mL. After the stability test, 8 mL electrolyte were taken out, and 2 mL 1 M H$_2$SO$_4$ were added into the electrolytes to dissolve possible insoluble Fe species. ICP-MS analysis was carried out to determine the Fe contents in electrolytes after durability tests. For comparison, 8 mL of fresh 0.1 M KOH and 2 mL 1 M H$_2$SO$_4$ were also mixed to detect Fe contents of fresh electrolytes.

A carbon paper integrated with a gas diffusion layer supported on nickel foam (Sigracet carbon paper, 28BC) was employed in the GDE half-cell tests. To obtain a homogeneous catalyst loading, 5 mg of as-prepared catalysts were dispersed into 0.5 mL ethanol containing, containing 10 μL of 5% wt. Nafion and followed by sonication for 1 h. Subsequently, 200 μL catalysts ink were drop-cast on GDE (1.5×1.5 cm$^2$). Then the GDE with catalysts was transferred into a vacuum oven at 40 °C. After fully drying for 30 min, the catalysts were evaluated on a home-made GDE configuration as shown in Fig. 4g. A saturated calomel electrode and graphite rod were used as referenced electrode and counter electrode, respectively. Sufficient pure oxygen was purged into the below chamber with a flow rate of 150 mL/min, 100% humidity and the temperature of the flowing electrolyte was controlled at 70 °C. Polarization curves of as-prepared catalysts were measured in 1 M KOH until stable activation CV curves were observed. The potentials were corrected to RHE with IRs compensation as described above.

The intrinsic activity of as-prepared samples was determined by MA and TOF. Given the atomically dispersed Fe single atoms and the porous feature of as-prepared Fe SACs, we assume that all the Fe sites act as accessible sites in the calculations. The MA of catalysts was calculated based on the following equations:

$$MA = \frac{I}{M} \quad (7)$$

Where $I$ is the reduction current ($A$) recorded in GDE measurements, $M$ is the meatal loading mass ($g$) of electrocatalysts. The metal contents were determined by ICP-MS.

The TOF of catalysts was obtained by following equations:

$$TOF = \frac{I}{4NF} \quad (8)$$

Where $I$ is the reduction current ($A$) recorded in GDE measurements, $N$ is the number of atomic Fe sites obtained from ICP-MS, and $F$ is the Faraday's constant. And $N$ of 20% Pt/C was determined by following

equations according to previous literature:

$$N = w_{Pt}*C_{cat}*D/M_{Pt} \quad (9)$$

Where $w_{Pt}$ is the Pt concentration of Pt/C, $C_{cat}$ is the mass loading of Pt/C catalyst on GDE, $D$ is the dispersion (26%), and $M_{Pt}$ is the mass per mole of Pt.

## Characterizations
XRD was detected on D8 advance (Bruker AXS corporation) using Cu Kα radiation. Raman spectra were recorded on a Thermo Fisher DXR instrument with a wavelength of 532 nm. The morphology and microstructure of as-prepare catalysts were characterized by TEM (JEM-2100F) and SEM (JSM-7500F). AFM was performed on Shimadzu SPM-9700. The electronic structure and valence state of as-prepared samples were detected by XPS on a VG ESCALABMK II instrument with Al Kα irradiation. The pore structure and specific surface area of catalysts were determined by N$_2$ adsorption and desorption isotherms on a 3FLEX instrument (Micromeritics Instrument Ltd.). The metal contents of catalysts were detected by using ICP-OES (Thermo Scientific ICAP 6300). XAFS were conducted at 1W1B station of the Beijing Synchrotron Radiation Facility. Operando XAFS measurements of the Fe K-edge were performed in fluorescence mode using a Lytle detector in the 12B2 Taiwan beamline at SP-8 (Japan). In situ Raman spectra were collected on a Thermo Fisher DXR2 microscope with 533 nm laser excitation in O$_2$-saturated electrolytes at room temperature. In situ ATR-SEIRAS experiments were collected with 4 cm$^{-1}$ resolution and at least 128 coadded scans using an FTIR spectrometer (Nicolet iS50, Thermo Scientific) equipped with a liquid nitrogen-cooled MCT detector. Electrochemical measurements were conducted using a Princeton potentiostat (Princeton Applied Research).

## Computational details
In order to understand the inter-site heterogeneity-induced effect of hierarchically porous Fe SAs-HP, the calculations on the ORR mechanisms were performed using DFT implemented in the Vienna Ab initio Simulation Package[50,51]. The exchange and correlation effects were treated using the generalized gradient approximation with the Perdew–Burke–Ernzer–hof functional[52]. The projector augmented-wave method was used to describe interactions between the core and valence electrons[53]. Plane waves were included for the electronic wave functions up to cutoff energy of 500 eV. Five structural models, including Fe-N$_4$ sites at microporous and mesoporous sites (Fe SAs-HP), Fe-N$_4$ sites anchoring at microporous sites (Fe SAs-MCP) or at mesoporous sites (Fe SAs-MSP), and conventional in-plane type Fe-N$_4$ sites were built and compared as shown in Supplementary Figs. 18–25. The Fe-N$_4$ sites were built by replacing six carbon atoms of graphene flake with one Fe and four N atoms anchoring at microporous or mesoporous edges. The distance between microporous and mesoporous Fe sites in Fe SAs-HP is 9.1 Å. Gaussian smearing of 0.08 eV was applied for the geometry optimization and total energy computations. The convergence criteria for the energy and force were set to $10^{-6}$ eV and −0.03 eV/Å, respectively. The reciprocal space was sampled by a grid of (1×1×1) $k$-points generated automatically using the Monkhorst–Pack method[54].

## Data availability
The data that support the findings of this study are available from the corresponding author upon request.

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

## Acknowledgements

This work was supported by Taishan Scholars Program of Shandong Province (no. tsqn201909065), the National Natural Science Foundation of China (no. 22108306), Shandong Provincial Natural Science Foundation (ZR2021YQ15), the Fundamental Research Funds for the Central Universities (22CX07009A), Hefei National Research Center for Physical Sciences at the Microscale (KF2021107), the State Key Laboratory of Organic-Inorganic Composites (oic-202101006).

## Author contributions

P.Z. and Y.P. designed the project, and wrote the manuscript. P.Z. prepared the catalyst samples, tested the ORR performance, characterized samples, analyzed the data, and wrote the corresponding sections. H.C. and P.Z. tested the in situ Raman experiments and operando EXAFS experiment and analyzed all the XANES and EXAFS data. H.Z., T.L. and P.Z. conducted DFT calculations, analyzed the calculation data, and wrote the corresponding section. K.C., Y.Z., J.L. and S.H. anticipated the preparation of catalysts. R.H. and W.Z. assembled and tested the zinc-air batteries and PEMFC. Y.P. and Y.L. supervised the project, directed the research, and established the final version of the manuscript. P.Z., H.C. and H.Z. contributed equally to this work.

## Competing interests

The authors declare no competing interests.
