## [Peer Review File · Nature Communications]

Inter-site structural heterogeneity induction of single atom Fe catalysts for robust oxygen reductionREVIEWER COMMENTS

Reviewer #1 (Remarks to the Author):

This work aims to develop efficient strategies to build hierarchically dispersed porous metal-nitrogen-carbon catalysts with asymmetric coordination environment to be utilized in oxygen reduction reaction (ORR). The authors report an efficient hierarchically porous Fe single atom catalyst (Fe SAs-HP) prepared with Fe atoms densely resided at micropores and mesopores. To assess the performance of Fe SAs-HP and monitor the structural evolution of catalyst, they carried out detailed experimental analysis including in situ Raman and operando XAS measurements combined with multilevel theoretical calculations. The paper clearly presents the synergetic effect between mesoporous and microporous sites by unraveling the structural origin for superior performance of Fe SAs-HP than single pore-sized single atom Fe catalysts. DFT calculations constructing prototype Fe-N₄ sites demonstrate both the individual and cooperative role of micro and mesoporous sites through 4e⁻ ORR mechanism. In that sense, it permits to establish structure-activity relationships by addressing the queries due to incomplete atomistic understanding of the ORR in relation to structural heterogeneity induction. Therefore, these results are important since they contribute to advance the growing field of electrocatalyst design using conductive materials with non-precious metals for both experimental and computational community.

Overall, the paper is well-written and discussed and the conclusions are consistent with the results. Concluding, I recommend the publication of this work in Nature Communications after minor revisions.

Additional Comments:

1. In Figure 4b, I suggest that the authors can present the free energy vs reaction pathway more clearly. As such, they can illustrate the key elementary steps such as O₂ binding, *OH intermediate formation and its desorption explicitly on the reaction pathway by labeling on reaction axis or so. That way, it might be easier for the reader to distinguish RDS for each prototype sites.
2. Regarding the dynamic response of the system to the solvent environment and surface charge effects, did authors check solid-liquid interactions? In reference 41, the authors show the role of axial H₂O co-adsorbed to the Fe center having implications for the thermodynamics and mechanism of ORR. Thus, I wonder if the authors can look at that through AIMD.

Reviewer #2 (Remarks to the Author):

This manuscript reports a Fe/N/C catalyst for the electrochemical oxygen reduction. Many of state-of-the-art in this class of catalysts are prepared by heat-treating Fe, C, N and Zn precursors, and this work also seems to follow that trend. Thus, this manuscript is meaningful if only the authors claim, "ultra-high mass activity", is reasonably validated. However, the authors' editing style makes it difficult. In Fig 3b and 3d, the authors compared E_{1/2} and E_{onset} against literatures, but this comparison is not fair because the amount of catalyst is not uniformed. The mass activity is discussed in Fig 3g and 3h, but this is not enough because this experimental setup is not very popular and any comparison against other earlier works are not provided. I cannot recommend this manuscript for publication.

Reviewer #3 (Remarks to the Author):

In this work, the authors prepared a hierarchically porous Fe single-atom catalyst with Fe loaded on the micropore and mesopore edges (Fe SAs-HP). Fe SAs-HP showed high catalytic activity for ORR, and the effects of the inter-site structural heterogeneity on the catalytic performance were studied in detail, including the dynamic switching behavior of the active sites in Fe SAs-HP. This study provides a rational example for the optimization of single-atom catalysts via pore structure engineering. However,

some key questions need to be resolved.

1. According to Supplementary Figure 1b, three catalysts all have abundant mesoporous structures. How does the author quantify this further?
2. What reaction process of the ORR for different catalysts according to the Tafel slope? Does a small Tafel slope indicate good catalytic performance?
3. How did pore structure affect catalytic stability? Why was the catalytic stability of a single pore structure so inferior? The authors should provide detailed explanations and corresponding experimental evidence.
4. The dynamic switching behavior of Fe single atoms in Fe SAs-HP had been studied. Did the active sites in the Fe SAs-MCP and Fe SAs-MSP have similar behavior?
5. According to this manuscript, the atomically dispersed Fe active sites could be shielded by SCN⁻. However, the Fe SAs-HP after adding KSCN solutions still displayed high activity for ORR. What is the real active site for Fe SAs-HP after adding KSCN solutions? How did the pore structure engineering affect the poisoning of SCN⁻ for Fe active sites?

Point-by-point response to the reviewers' comments

We sincerely thank the editor and reviewers for their valuable comments and suggestions on our manuscript “**inter-site structural heterogeneity induction of single atom Fe catalysts for robust oxygen reduction with high mass activity**” (NCOMMS-23-44547-T), which certainly helped to improve the quality of our manuscript. We have incorporated the suggestions made by reviewers. All the changes have been highlighted in **blue** in the Revised Manuscript with changes marked-up, and the point-by-point responses are presented below.

Reviewer #1:

Comments: This work aims to develop efficient strategies to build hierarchically dispersed porous metal-nitrogen-carbon catalysts with asymmetric coordination environment to be utilized in oxygen reduction reaction (ORR). The authors report an efficient hierarchically porous Fe single atom catalyst (Fe SAs-HP) prepared with Fe atoms densely resided at micropores and mesopores. To assess the performance of Fe SAs-HP and monitor the structural evolution of catalyst, they carried out detailed experimental analysis including *in situ* Raman and *operando* XAS measurements combined with multilevel theoretical calculations. The paper clearly presents the synergetic effect between mesoporous and microporous sites by unraveling the structural origin for superior performance of Fe SAs-HP than single pore-sized single atom Fe catalysts. DFT calculations constructing prototype Fe-N₄ sites demonstrate both the individual and cooperative role of micro and mesoporous sites through 4e-ORR mechanism. In that sense, it permits to establish structure-activity relationships by addressing the queries due to incomplete atomistic understanding of the ORR in relation to structural heterogeneity induction. Therefore, these results are important since they contribute to advance the growing field of electrocatalyst design using conductive materials with non-precious metals for both experimental and computational community. Overall, the paper is well-written and discussed and the conclusions are consistent with the results. Concluding, I recommend the publication of this work in Nature Communications after minor revisions.

Additional Comments:

1. In Figure 4b, I suggest that the authors can present the free energy vs reaction pathway more clearly. As such, they can illustrate the key elementary steps such as O_2 binding, $*OH$ intermediate formation and its desorption explicitly on the reaction pathway by labeling on reaction axis or so. That way, it might be easier for the reader to distinguish RDS for each prototype sites.

Answer: Thank you very much for your valuable suggestions. We have corrected the styles and clearly presented the free energy of key steps vs reaction pathway in Figure 4b. Additionally, the RDS for each site has been highlighted and marked with star symbol to make it clear to the readers.

Fig. 4 (b) Free energy against reaction pathway of as-prepared catalysts.

2. Regarding the dynamic response of the system to the solvent environment and surface charge effects, did authors check solid-liquid interactions? In reference 41, the authors show the role of axial H_2O co-adsorbed to the Fe center having implications for the thermodynamics and mechanism of ORR. Thus, I wonder if the authors can look at that through AIMD.

Answer: Thank you for your valuable comments and suggestions. An explicit solvent model has been implemented to assess solid-liquid interactions by introducing OH^- and H_2O molecules as shown in supplementary Fig. 31. In addition, since all AIMD simulations are performed under constant charge conditions, the charge extrapolation

method developed by Chan and Nørskov is applied for constant potential corrections. (*J. Phys. Chem. Lett.* **6**, 2663–2668 (2015); *J. Phys. Chem. Lett.* **7**, 1686–1690 (2016).). A production period of 8 ps was used to evaluate the solid-liquid interaction between active sites and solvents. The initial and final configuration of the 8 ps AIMD simulation on mesoporous sites (Fe SAs-HP@ MSP) and adjacent microporous sites (Fe SAs-HP@MCP) were shown in **Fig. 4d** and **4e**. The mesoporous Fe sites (Fe SAs-HP@ MSP) preferentially adsorbed O₂ molecules as exhibited in **Fig. 4d** and **Fig. 4f**, indicating a favorable oxygen activation steps and was also consistent with reaction free energies in Fig. 4b. Different from the mesoporous Fe sites, H₂O molecules tended to be adsorbed on microporous Fe sites during the production period due to the shortened Fe-O distance between Fe sites and H₂O molecules as shown in **Fig. 4e** and **Fig. 4g**, demonstrating an inferior ORR dynamic and O₂ activation process on microporous Fe sites. The energy and temperature fluctuated within a certain range, suggesting the structure of dual Fe sites system is stable during ORR process as shown in supplementary Fig. 32, suggesting the structure of dual Fe sites system is stable during ORR process. The Fe-N₄ site on mesoporous edge in the dual site system behaved more active and was deemed as active sites during ORR process.

Fig. 4 Theoretical calculations of as-prepared catalysts. (d) and (e) initial and final

configuration of the 8 ps AIMD simulations for mesoporous Fe sites (Fe SAs-HP@MSP) and microporous Fe sites (Fe SAs-HP@MCP), respectively. (f) and (g) the distance between Fe sites and O atom in adsorbed H₂O (green) and O₂ (red) as a function of time of AIMD simulations at 298 K.

Supplementary Figure 31. AIMD simulated interfacial structure of Fe SAs-HP. (a) initial state and (b) final state of dynamic adsorption.

Supplementary Figure 32. Energy and temperature fluctuation during simulations.

Reviewer #2:

Comments: This manuscript reports a Fe/N/C catalyst for the electrochemical oxygen reduction. Many of state-of-the-art in this class of catalysts are prepared by

heat-treating Fe, C, N and Zn precursors, and this work also seems to follow that trend. Thus, this manuscript is meaningful if only the authors claim, "ultra-high mass activity", is reasonably validated. However, the authors' editing style makes it difficult. In Fig 3b and 3d, the authors compared $E_{1/2}$ and E_{onset} against literatures, but this comparison is not fair because the amount of catalyst is not uniformed. The mass activity is discussed in Fig 3g and 3h, but this is not enough because this experimental setup is not very popular and any comparison against other earlier works are not provided. I cannot recommend this manuscript for publication.

Response to the reviewer: We thank the reviewer for your valuable comments and concerns about our work. We also appreciate the doubts and efforts of the reviewer, which certainly contributes to improve the quality our manuscript. We have responded the questions raised by the reviewer in the hope of gaining the reviewer's approval. The detailed responses were given point by point as listed below. Thank you again for your detailed comments on our work.

1. Response to the novelty of preparation method for Fe/N/C in this manuscript.

Response: We noticed the preparation method of Fe/N/C reported recently. And most reported Fe/N/C single atom catalysts were obtained from well-defined metal organic framework (MOF, such as ZIF-8) by heat-treating. The Zn would evaporate under high temperature and form micropores. Nevertheless, controllable pore engineering and Fe-N₄ sites are hardly achieved. **More importantly, the real active sites and interactions between hierarchical porous Fe sites were rarely disclosed.** The relationship between activity and stability with pore structure (sites) was still unclear. In these similar preparations, several unfavorable scenarios would arise.

1) Firstly, it was difficult to regulate the pore structure of these derivatives due to rigid coordination structure even though SACs with hierarchical pores were reported to have remarkable ORR activity.

2) The Fe atoms were generally anchored by four atoms, forming in-plane type Fe SACs, due to rigid coordination structure provided by MOF. The symmetric Fe-N₄ sites would lead to strong adsorption of oxygen and unsatisfactory activity.

3) The MOF-derivatives always suffered metal aggregations during pyrolysis. Tedious acid washing process was needed to obtain isolated atoms catalysts. More importantly, the active sites were buried into the bulk, making it difficult for the reactants to access active centers.

Given that, we developed an efficient and controllable strategy to prepare hierarchically porous Fe SAs-HP through molecular self-assembly due to π - π stacking of polycyclic aromatic hydrocarbons and heme chloride macrocycle without tedious acid washing process; The novelty of this strategy was listed as follows:

- (1) The pore structure could be adjusted easily by incorporating soft templates ZnO and NH_3/N_2 atmosphere. **Benefiting from the unique ZnO templates and polycyclic aromatic hydrocarbons substrates, the Fe atoms were then found to be preferentially anchored at micropore and mesopore edges;**
- (2) Thus, it allowed us to correlate the relationship between site structure and performance at atomic level. Impressively, **an inter-site structural heterogeneity induced optimization of Fe single atoms were systematically revealed** at atomic precise owing to the excellent platforms of Fe SAs-HP and advanced characterizations; An explicit solvent model has been implemented to assess solid-liquid interactions by introducing OH^- and H_2O molecules as shown in Fig. 4d-4e and supplementary Fig. 31. **The mesoporous Fe-N₄ sites adjacent to microporous Fe-N₄ sites were verified as real active centers due to electronic regulations of adjacent microporous Fe-N₄ sites.**
- (3) The relationship between stability and porous structure was further disclosed in combination with *operando* XAFS and Raman (supplementary 39-42). The single microporous Fe-N₄ sites tended to aggregate into FeO species under high overpotentials. The single mesoporous Fe-N₄ tended to leach out from N₄ plane. The unfavorable switching behaviors of single porous Fe sites caused the loss of durability. Neither aggregation or obvious leaching of hierarchically porous FeN₄ sites can be observed for Fe SAs-HP, suggesting the strong interactions between adjacent microporous and mesoporous Fe sites were conducive to stabilize Fe atoms.

Above discussion on AIMD simulations and the relationship between stability and site structure have been supplemented in the main text in the part of “Theoretical evidence of structural heterogeneity induction effect” and “Investigation of dynamic mechanisms on active Fe-N₄ sites”.

Response Figure 1. (a) and (b) *Operando* XANES of Fe K-edge and corresponding FT-EXAFS fitting analysis for Fe SAs-MCP under 0.1 M KOH.

Response Figure 2. (a) and (b) *Operando* XANES of Fe K-edge and corresponding FT-EXAFS fitting analysis for Fe SAs-MSP under 0.1 M KOH.

2. Response to “the comparison of $E_{1/2}$ and E_{onset} against literatures is unfair due to the catalyst is not uniformed.”

Answer: We thank the reviewer’s valuable comments and concerns about the comparison method. The $E_{1/2}$ and E_{onset} potentials obtained from rotate disk electrode (RDE, with a standard diameter of 5 mm) were important parameters to assess

apparent activity of as-prepared catalysts and were generally employed to compare the apparent activity of catalysts (Nat. Commun. 2020, 11, 3049; J. Am. Chem. Soc. 2022, 144, 2197-2207; J. Am. Chem. Soc. 2023, DOI:10.1021/jacs.3c08556).

Thus, the catalysts loadings were always optimized to **gain optimum $E_{1/2}$ and E_{onset} regardless the loadings of non-noble-based catalysts**. Moreover, $E_{1/2}$ and E_{onset} potentials were hardly enhanced by further adjusting the loading of catalysts due to limited intrinsic activity. The same standard was also used in evaluating counterparts and Pt/C benchmarks. Therefore, $E_{1/2}$ and E_{onset} were employed to compare the apparent activity of as-prepared catalysts. In addition, the activity normalized by Fe contents (mass activity) and TOF jointly demonstrated the high ORR activity of hierarchically porous Fe sites. Thank you for your concern about catalyst loadings. We have listed catalyst loadings of compared catalysts in reported literatures to give a clear view. According to the summarized apparent ORR activity, $E_{1/2}$ and E_{onset} showed weak correlations with the catalyst loadings. **We have supplemented the catalyst loadings into supplementary Table 3 and 4 to provide clarity for the readers.**

Response Table 1. Alkaline ORR performance and peak power density of as-assembled zinc-air batteries (ZAB) of recent reported Catalysts.

Catalyst	E_{onset} (V vs. RHE)	$E_{1/2}$ (V vs. RHE)	Peak power density of ZABs (mW cm^{-2})	Catalyst loadings (mg cm^{-2} , RDE)	Reference
Fe SAs-HP	1.06	0.94	254.2	1.0	This work
Fe-N/P-C-700	0.94	0.87	133.2	0.6	J. Am. Chem. Soc. 142 , 2404–2412 (2020)
Fe _H -N-C	-	0.91	225	0.6	Adv. Mater. 35 , 2210714 (2023)
FeN ₃ OS	1.01	0.874	-	0.4	Angew. Chem. Int. Ed. 60 , 25296–25301 (2021)
Fe ₁ Se ₁ -NC	1.0	0.88	-	0.2	Appl. Catal. B-Environ. 308 , 121206

Fe/Meso-NC-1000	0.97	0.885	188.4	0.3	(2022) Adv. Mater. 34 , 2107291
Fe-N-GDY	1.05	0.89	249	0.4	(2022) Angew. Chem. Int. Ed. 61 , e202208238
OAC	0.98	0.85	113	0.3	(2022) Appl. Catal. B-Environ. 305, 121058
Fe-SA-NSFC	1.01	0.91	247.7	0.5	(2022) Nat. Commun. 11 , 5892
Fe SAs-Fe ₂ P NPs/NPCFs-2.5	1.03	0.91	236	0.5	(2020) Adv. Mater. 34 , 2203621
Co ₂ /Fe-N@CHC	1.03	0.915	232.4	0.3	(2022) Adv. Mater. 33 , 2104718
Fe,Mn/N-C	0.979	0.928	160.8	0.1	(2021) Nat. Commun. 12 , 1734
Fe,P-DAS@MPC	1.02	0.92	230	0.255	(2021) Adv. Energy Mater. 13 , 2203611
					(2022)

Response Table 2. Acidic ORR performance of as-assembled zinc-air batteries (ZAB) of recent reported Catalysts.

Catalyst	E _{onset} (V vs. RHE)	E _{1/2} (V vs. RHE)	Catalyst loadings (mg cm ⁻² , RDE)	Reference
Fe SAs-HP	0.90	0.78	1.0	This work
Fe ₁ Se ₁ -NC	0.88	0.74	0.2	Appl. Catal. B: Environ. 308 , 121206 (2022).
OAC	0.86	0.71	0.3	Appl. Catal. B: Environ. 305 , 121058 (2022).

Fe/OES	0.80	0.71	0.4	Angewandte Chemie. 132 , 7454-7459 (2020).
CoFe@C	0.80	0.70	0.408	Angewandte Chemie. 131 , 1997-2001 (2019).
Fe/N-CNRs	0.89	0.73	0.4	Adv. Funct. Mater. 31 , 2008085 (2021)
Fe/Ni-N-PCS DM-SAC	0.87	0.71	0.255	J. Colloid Interf. Sci. 633 , 828-835 (2023).
Fe ₅₀ -N-C-900	0.88	0.73	0.1	Small. 14 , 1703118 (2018).
Fe-N-C/N OMC	0.91	0.73	0.3	Appl. Catal. B: Environ. 280 , 119411 (2021).
FeCu-DA/NC	0.89	0.78	0.5	J. Mater. Chem. A. 8 , 16994-17001 (2020).

3. Response to “The mass activity is discussed in Fig 3g and 3h, but this is not enough because this experimental setup is not very popular and any comparison against other earlier works are not provided”

Answer: We thank the reviewer’s valuable comments and concerns about experimental setups and mass activity. Due to limited evaluation setups and insufficient preparation method, the relationship between catalyst structure and intrinsic performance is rarely reported. Even though the SACs with hierarchically porous structure were deemed as efficient geometry for ORR. Here, the gas diffusion electrode (GDE) was employed to access the intrinsic activity of Fe sites and contribute to construct the relationship between site structure and performance. (*Nat. Catal.* **4**, 615-622 (2021))

Generally, the ORR took place on the active Fe sites. The reactants should access the active centers first. Unfortunately, this process was severely affected by pore structure of as-prepared catalysts, leading to a mass-transport-controlled process. Besides, the reactants concentration will also determine the reactive activity, namely the recorded current signals. The conventional rotate disk electrode tests could not satisfy the above

requirements due to **the low oxygen solubility and mass-transport-controlled process** of different pore-structure of as-prepared catalysts. **Therefore, conventional RDE tests cannot establish scientific and convincing relationship between Fe sites structure and performance at atomic precise.** Consequently, GDE tests were employed to evaluate the intrinsic activity of Fe sites as evidenced by Fig. 3f. **The GDE tests were developed to assess intrinsic activity of efficient sites with different geometrics.**

Mass activity and TOF were obtained to evaluate the site activity of as-prepared catalysts, which showed a pore-structure-dependent correlation. And we added the comparison of mass activity and TOF to other earlier work in Response table 4. Fe SAs-HP exhibited remarkable site activity and mass activity, even under GDE test system (*Nat. Catal.* **4**, 615-622 (2021)).

To verify the practicality of GDE method in assessing intrinsic activity, the turnover frequency was also obtained according to the in situ NaNO₂ poisoning method (NPM) developed by Kucernak et al. (*Nat. Commun.* 2016, 7, 13285). The TOF at 0.85 V estimated for Fe SAs-HP was 3.7 s⁻¹, which was 2.3 and 9.3 times higher than those of single microporous Fe-N₄ sites (Fe SAs-MCP, 1.57 s⁻¹) and single mesoporous Fe-N₄ sites (Fe SAs-MSP, 0.39 s⁻¹). **These results were basically consistent with GDE methods as summarized in Response Table 3.**

The TOF results indicated that GDE tests were also acceptable to assess intrinsic activity. **However, TOF evaluated by NPM expanded the gap between Fe SAs-HP and Fe SAs-MCP. Considering the geometry and multilevel pore size, which will hinder the diffusion of NO₂⁻, GDE method was adopted to assess the intrinsic activity.** We then compared TOF and mass activity of Fe SAs-HP with reported catalysts as shown in Response Table 4. Fe SAs-HP exhibited remarkable ORR activity with high mass activity of 4.14 x 10⁴ A g_{Fe}⁻¹. **For the sake of rigor and to highlight the significance of inter-site structural heterogeneity induced effect, the title “ultra-high mass activity” was revised to “high mass activity”.** The comparison of TOF and mass activity for recently reported works has been supplemented in

supplementary Table 5.

Response Figure 3. Determination of TOF for as-prepared catalysts through nitrite poisoning. Left column, LSV curves before, during and after nitrite adsorption in a 0.5 M acetate buffer at pH 5.2. Right column, CV curves before and during nitrite adsorption in the nitrite reductive stripping region. Catalysts loadings and nitrite concentration were kept at 0.27 mg cm⁻² and 125 mM, respectively.

Response Table 3. Comparison of TOF at 0.85 V determined by GDE and in situ nitrite poisoning method (NPM).

Catalysts	TOF @ 0.85V (GDE, s ⁻¹)	TOF @ 0.85V (NPM, s ⁻¹)
Fe SAs-HP	2.40	3.70
Fe SAs-MCP	1.07	1.57
Fe SAs-MSP	0.99	0.39

Response Table 4. Comparison of TOF and mass activity at 0.8 V for Fe SAs-HP and reported catalysts.

catalysts	TOF (s ⁻¹)	Mass activity (A g _{Fe} ⁻¹)	reference
Fe SAs-HP	5.99	4.14 x 10 ⁴	This work
Fe SACs	4.3	1.5 x 10 ³ (GDE)	Nat. Catal. 4 , 615-622 (2021)
Cyan-Fe-N-C	0.79	1.142 x 10 ³	Adv. Mater. 35 , 2305945 (2023)
TAP 900@Fe	0.087	4.0	Adv. Mater. 35 , 2211022 (2023)
sur-FeN ₄ -HPC	1.01	16.5	Energy Environ. Sci. 15 , 2619 (2022)
TPI@Z8(SiO ₂)-650-C	1.63	-	Nat. Catal. 2 , 259-268 (2019)
Fe-SA-NSFC	0.22	-	Nat Commun. 11 , 5892 (2020)
LTHT-FeP aerogel	0.25	-	Angew. Chem. Int. Ed. 59 , 2483-2489 (2020)

Additionally, we also added *operando* XAS of comparative samples (Fe SAs-MCP and Fe SAs-MSP) and revealed the **inter-site structural heterogeneity induced optimization of Fe single atoms with enhanced activity and stability**. The

interactions between activity and stability with structure of catalysts were both disclosed, which will facilitate rational regulation of efficient Fe SACs. We hope our efforts may gain the reviewer's approval and appreciate the valuable comments to improve the quality of our manuscript.

Reviewer #3:

Comments: In this work, the authors prepared a hierarchically porous Fe single-atom catalyst with Fe loaded on the micropore and mesopore edges (Fe SAs-HP). Fe SAs-HP showed high catalytic activity for ORR, and the effects of the inter-site structural heterogeneity on the catalytic performance were studied in detail, including the dynamic switching behavior of the active sites in Fe SAs-HP. This study provides a rational example for the optimization of single-atom catalysts via pore structure engineering. However, some key questions need to be resolved.

1. According to Supplementary Figure 1b, three catalysts all have abundant mesoporous structures. How does the author quantify this further?

Response: We thank the reviewer for the valuable comments and discussions, which should be better discussed in the manuscript. In supplementary Fig. 1b, the pore size centred at ~32nm can be observed for Fe SAs-HP and Fe SAs-MSP due to porogenesis of ZnO soft templates, which was consistent with TEM observations in Fig. 1b and supplementary Fig. 5. While intensive pore distributions can also be observed in Fe SAs-MCP, Fe SAs MSP and Fe SAs-HP below 30 nm.

With respect to Fe SAs-MCP, the N₂ adsorption/desorption curves were well consistent with type IV and H₄ hysteresis loops according to the IUPAC classification. The flat H₄ hysteresis loops suggested that the parallel slit-type pores were formed due to the stacking of carbon nanosheets, **which would not contribute to the anchoring of active Fe atoms owing to the large slit pore size (from 2 nm to 30 nm).** The N₂ adsorption/desorption results were also in good agreement with TEM observations (Supplementary figure 5), where carbon layers were stacked for Fe SAs-MCP. **Therefore, it would be reasonable to exclude contributions of this part of mesopore size in Fe SAs-MCP when correlating the relationship between site activity and**

site structure (microporous and mesoporous Fe sites) on a specific carbon layer. The contribution of slit-type pores to total specific surface area (312.7 m²/g) was 42.6 m²/g, which take relative low proportions (13.6%) to the total surface area.

As for Fe SAs-MSP, similar to Fe SAs-MCP, the N₂ adsorption/desorption curves were also well consistent with type IV and H₄ hysteresis loops according to the IUPAC classification. However, a broad peak of pore size distribution centred at 0.8 nm with low intensities can be observed, which was quite different from those for Fe SAs-MCP and Fe SAs-MSP, centred at 0.47 nm, as shown in supplementary figure 1c. These micropores with broad pore size distribution were likely ascribed to the stacking of slit layers (micropores part at the end to two layers), since the Fe SAs-MSP possessed the ordered structure with lowest defects as verified by EPR tests in figure 1k. **Therefore, it would be also reasonable to exclude contributions of this part of micropore in Fe SAs-MSP when correlating the relationship between site activity and site structure (microporous and mesoporous Fe sites) on a specific carbon layer.**

The specific surface areas of mesopores (S_{msp}) of Fe SAs-HP, Fe SAs-MCP and Fe SAs-MSP were added into supplementary Table 1. The mesopore area were obtained by exclude the micropore area in total BET surface area. Considering the relative low proportions of slit-type pores of Fe SAs-MCP and Fe SAs-MSP and the key role of microporous sites in regulating the catalytic performance, the proportion of S_{mcp} ($S_{\text{mcp}}/S_{\text{BET}}$) was applied as an indicator to quantify the effect of pore size. According to the key indicator of $S_{\text{mcp}}/S_{\text{BET}}$, Fe single atom catalysts with hierarchically porous sites (Fe SAs-HP) exhibited remarkable ORR activity.

Response Table 1. BET surface area, micropore surface area (S_{mcp}) and mesopore area (S_{msp}) of as-prepared catalysts.

Catalysts	S_{BET}	S_{mic}	S_{msp}	$S_{\text{mic}}/S_{\text{BET}}$
Fe SAs-HP	578.9	344.5	234.4	59.5%
Fe SAs-MCP	312.7	270.1	42.6	86.4%
Fe SAs-MSP	137.9	54.1	83.8	39.2%

Above discussions were also supplemented in the supporting information and main

text in the manuscript.

2. What reaction process of the ORR for different catalysts according to the Tafel slope?

Does a small Tafel slope indicate good catalytic performance?

Response: We thank the reviewer for the valuable discussions, deepening the understanding of Tafel slope. The 4-electron ORR process was much more sophisticated due to complex oxygenated intermediates (*O, *OH, *OOH and *O₂) and multiple reaction steps, unlike the two -electron hydrogen evolution reaction, where the reaction process (Volmer-Heyrovsky or Volmer-Tafel process) can be easily obtained by Tafel slope. A commonly suggested reaction process for ORR on Fe-N-C in alkaline electrolyte has the following steps: *+O₂+H₂O+e⁻→*OOH+OH⁻, *OOH+e⁻→*O+OH⁻, *O+H₂O+e⁻→*OH+OH⁻, *OH+e⁻→OH⁻. Experimental determination of the exact ORR process according to Tafel slope yet is still very challenging and the rate determination step (RDS) of ORR for FeNC is also controversial as each one or two of four-step process above can determine the ORR process (*J. Am. Chem. Soc.* 2023. DOI:10.1021/jacs.3c09193).

The Tafel slopes of as-prepared catalysts were 84.0, 85.1 and 95.7 mV dec⁻¹ for Fe SAs-HP, Fe SAs-MCP and Fe SAs-MSP, respectively. **According to the analysis of an ideal Tafel slope, the RDS for Fe SAs-HP, Fe SAs-MCP and Fe SAs-MSP can be assigned to the first step of *+O₂+H₂O+e⁻→*OOH+OH⁻, namely activation of O₂** (*J. Electrochem. Soc.* 2012, 159(11), H864-H870, DOI:10.1149/2.022211jes).

We also noticed that these results were partly consistent with our DFT calculations in Figure 4b. The Tafel analysis was in good agreement with Fe SAs-MCP where the RDS was the activation of O₂ on microporous Fe sites. In general, DFT calculations give the free energy of ORR process on a specific site. As for Fe SAs-HP, this discrepancy might be ascribed to the fact that the hybrid active sites were existed in Fe SAs-HP. Therefore, Fe SAs-HP exhibited an apparent RDS of oxygen activation. Benefitting from the interactions of inter Fe sites, Fe SAs-HP exhibited faster ORR kinetics than Fe SAs-MCP and Fe SAs-MSP. Note that, the analysis of RDS based on Tafel slope theory depends on the number of free sites and changes with changes in the

coverage of any site blocking species. As for Fe SAs-MSP, the RDS obtained from DFT calculations were the desorption of OH⁻, which was inconsistent with Tafel slope analysis. This reason might be ascribed to the low active sites and strong adsorption of *OH intermediates, which decreased the reaction kinetics and exhibited a low apparent Tafel slope.

In general, a smaller Tafel slope indicates a faster ORR kinetics as it represents a lower potential was needed to further improve the current density. (*J. Am. Chem. Soc.* 2022, 144, 15999–16005; *Nat. Commun.* 2020, 11, 4173.). The relevant literatures (ref.36 and ref. 37) were cited in the main text to explain the faster ORR kinetics of Fe SAs-HP.

3. How did pore structure affect catalytic stability? Why was the catalytic stability of a single pore structure so inferior? The authors should provide detailed explanations and corresponding experimental evidence.

Response: We sincerely thank the reviewer for the valuable discussion and comments. A harsh stability test was performed by loading ink of as-prepared catalysts on carbon paper (1 cm², Sigracet carbon paper, 28BC) at 0.2 V vs. RHE with high overpotentials for 8 h in 0.1 M KOH under standard three electrodes system. The Fe loadings and the volume of electrolytes were kept same. Fe loadings was kept at 0.0145 mg cm⁻² based on ICP-MS result and the volume of electrolyte was kept for 20 mL. Fe SAs-HP exhibited decent stability with high current retention of 93.5% and followed by Fe SAs-MCP (63.2%) and Fe SAs-MSP (58.1%), as shown in response Fig. 1. The dramatic fluctuation of stability curves was ascribed to the limited volume electrolytes. The carbon papers (catalysts) would be vulnerable to O₂ bubbles. After the stability test, 8 mL electrolyte were taken out and 2 mL 1 M H₂SO₄ were added into the electrolytes to dissolve possible insoluble Fe species (such as Fe(OH)₃, Fe₂O₃, FeO, etc.). ICP-MS analysis was carried out to determine the Fe contents in electrolytes after durability tests. For comparison, 8 mL of fresh 0.1 M KOH and 2 mL 1 M H₂SO₄ were also mixed to detect Fe contents of fresh electrolytes. The Fe contents were listed in Response Table 2. Carbon paper coated with catalysts after stability tests was sonicated in ethanol. And

The catalyst solutions were characterized by high resolution TEM and corresponding EDS mappings.

A higher Fe content in electrolytes was found for Fe SAs-MSP after long-term durability tests under rigid ORR conditions, demonstrating that the Fe atoms have been leached out from mesoporous Fe-N₄ sites. **The insufficient active sites would lead to great current loss during long-term durability tests.** A relative low Fe content can be observed for Fe SAs-HP, which could be ascribed to the strong interactions between adjacent microporous and mesoporous Fe-N₄ sites. The strong electron coupling effects of adjacent microporous (as evidenced by Bader charge analysis) might help to stabilize active Fe centers. And less Fe contents of Fe SAs-MCP were also observed, indicating the Fe atoms at microporous Fe-N₄ sites would not dissolve into electrolytes.

High-resolution TEM and corresponding EDS mapping were performed to further reveal the possible changes of microstructure and element states for as-prepared catalysts after rigid durability tests. Fe SAs-HP reserved the porous nature after durability tests and no obvious aggregated species can be observed in the whole randomly selected region in Response Figure 2a and 2b. In combination with ICP-MS results of Fe contents in electrolytes and *operando* XAFS, Fe atoms in Fe SAs-HP would be stable under harsh conditions. Only a small number of Fe atoms leached out from Fe-N₄ active sites and dissolved into electrolytes. As for single microporous Fe-N₄ sites (Fe SAs-MCP), no clear aggregation of Fe species can be observed from high resolution TEM due to the disturb of organic solvent such as nafion solutions. However, in the field of HAADF TEM images, **numerous tiny clusters can be observed throughout the regions as shown in Response Figure 2d and corresponding Fe mapping**, suggesting that the individual microporous Fe-N₄ sites tend to aggregate after long-term durability tests. **The aggregated Fe species reduced the number of active Fe sites in Fe SAs-MCP, thereby lowering the current density.** Most of the escaped Fe atoms would aggregate into tiny clusters rather than dissolve into electrolytes, which might be ascribed to the high Fe contents of Fe SAs-MCP (supplementary Table 1, 0.4299%). In the case of Fe SAs-MSP, no obvious aggregation of Fe species can be

observed in both TEM and HAADF TEM fields (response Figure 2e and 2f).

Actually, these results were also consistent with dynamic evolution of Fe SAs-MCP and Fe SAs-MSP revealed by operando XAFS results, which will be discussed in detail in Response 4.

Response Figure 1. Rigid stability tests of as-prepared catalysts at 0.2 V vs. RHE in 0.1 M KOH.

Response Figure 2. High resolution TEM, HAADF STEM and corresponding element mapping after harsh durability tests. (a) and (b) Fe SAs-HP; (c) and (d) Fe SAs-MCP; (e) and (f) Fe SAs-MSP.

Response Table 2. The Fe contents determined by ICP-MS of electrolytes after stability tests for as-prepared catalysts.

Samples	Fe contents in electrolytes (mg/L)
Fresh electrolytes	0.0651
Fe SAs-HP	0.1628
Fe SAs-MCP	0.1793
Fe SAs-MSP	0.2673

In conclusion, the individual microporous Fe sites tend to disengage from Fe-N₄ coordination structure and aggregated into tiny clusters during long-term durability test, which lead to the decreased stability. In the case of single mesoporous Fe sites, the Fe

sites tends to disengage from Fe-N₄ coordination structure and transferred into electrolytes solutions. Therefore, great current loss can be observed for Fe SAs-MSP. Different from scenarios of individual porous Fe sites, no obvious aggregation of Fe species and slight leach of Fe atoms can be observed for Fe SAs-HP, which lead to mild current loss during rigid durability tests. The results further confirmed that the strong interaction between adjacent microporous and mesoporous Fe sites was conducive to stabilize the Fe atoms.

4. The dynamic switching behavior of Fe single atoms in Fe SAs-HP had been studied. Did the active sites in the Fe SAs-MCP and Fe SAs-MSP have similar behavior?

Response: We sincerely thank the reviewer for the valuable suggestions. *Operando* XAS and *in situ* Raman for Fe SAs-MCP and Fe SAs-MSP were performed to monitor the dynamic switching behavior of active Fe sites. Interestingly, the dynamic switching behaviors of microporous and mesoporous Fe sites were quite different from each other. With the increasing of applied overpotentials in alkaline environment, the Fe-N coordination number (CN) of Fe SAs-CP was monitored to decrease from 3.8 at open circuit voltage (OCV) to 1.8 at higher overpotentials of 0.4 V as summarized in response table 3, indicating an evolution of Fe-N₄ to Fe-N₂. Meanwhile, the CN of adsorbed intermediates on microporous Fe sites (Fe-O¹) was observed to increase from 0 to 0.5 at 0.6 V. Impressively, new Fe-Fe scattering path can be surveilled at around 2.5 Å under 0.4 V as shown in response Figure 3b, suggesting the scattering path of FeO. The CN of Fe-Fe and the Fe-O² (lattice oxygen) was 6.1 and 3.6, respectively. Note that the standard CN for Fe-O and Fe-Fe was 6 and 12, respectively. The reduced CN for FeO could be ascribed to the formation of tiny cluster (Phys. Chem. Chem. Phys., 14, 11457–11467 (2012)). The results indicated that the microporous Fe-N₄ sites tended to break Fe-N bond and aggregate into FeO clusters under higher overpotentials, which accounted for their loss of current density during long-term durability tests. In addition, the intensity ratios of D-G band (I_D/I_G) for Fe SAs-MCP were observed to incline to higher values with the increase of overpotentials, demonstrating a dislocation of carbon matrix as shown in Response Fig. 4. In the case of Fe SAs-MSP, the dynamic

structural evolutions were surveilled to undergo a transformation from Fe-N₄ to Fe-N₂(OH) with the increasing of applied overpotentials from 1.0 V to 0.4 V as exhibited in response table 4. The dramatically structure deformation was also verified by in situ Raman results due to the increased I_D/I_G as shown in Response Fig. 6. The unsaturated Fe atoms were vulnerable to oxygenated intermediates and might leach out from N₄ plane, which would cause great loss of current density during durability tests.

Above discussions have been supplemented in the part of “Investigation of dynamic mechanisms on active Fe-N₄ sites” in the manuscript and corresponding supplementary information (supplementary Fig. 39-42, Table 8-9)

Response Figure 3. (a) and (b) *Operando* XANES of Fe K-edge and corresponding FT-EXAFS fitting analysis for Fe SAs-MCP under 0.1 M KOH.

Response Figure 4. (a) and (b) *In situ* Raman spectra of Fe SAs-MCP tested in 0.1 M KOH and corresponding I_D/I_G values with applied overpotentials.

Response Figure 5. (a) and (b) *Operando* XANES of Fe K-edge and corresponding FT-EXAFS fitting analysis for Fe SAs-MSP under 0.1 M KOH.

Response Table 3. *Operando* XAS analysis parameters of coordination number (CN) for active Fe sites of Fe SAs-MCP in 0.1 M KOH under working conditions.

Potentials	path	CN	R (Å)	dE	dW
dry	Fe-N	3.8(2)	1.94(3)	-7.5(3)	0.0084(7)
ocv	Fe-N	3.8(2)	1.95(2)	-8.3(3)	0.0083(7)
1.0	Fe-N	3.7(4)	1.94(3)	-7.9(4)	0.0084(6)
	Fe-O	-	-	-	-
0.8	Fe-N	3.7(4)	1.95(5)	-8.6(5)	0.0081(5)
	Fe-O	0.3(4)	2.02(4)	-5.2(7)	0.0097(7)
0.6	Fe-N	3.2(4)	1.96(5)	-11.4(5)	0.0092(7)
	Fe-O	0.5(3)	2.03(3)	-5.7(5)	0.0095(6)
0.4	Fe-N	1.8(2)	1.94(4)	-10.7(6)	0.0089(7)
	Fe-O ¹	0.3(4)	2.03(3)	-4.5(5)	0.0079(7)
	Fe-O ²	3.6(2)	2.13(3)	6.2(5)	0.0062(8)
	Fe-Fe	6.1(2)	3.08(4)	-1.8(7)	0.0095(7)

Response Table 4. *Operando* XAS analysis parameters of coordination number (CN) for active Fe sites of Fe SAs-MSP in 0.1 M KOH under working conditions.

Potentials	path	CN	R (Å)	dE	dW
dry	Fe-N	3.5(4)	1.96(2)	-12.3(4)	0.0105(5)
ocv	Fe-N	3.5(3)	1.95(2)	-10.2(5)	0.0106(4)
1.0	Fe-N	3.5(4)	1.96(4)	-11.5(5)	0.0106(7)
	Fe-O	-	-	-	-
0.8	Fe-N	3.4(4)	1.96(5)	-12.5(5)	0.0111(5)
	Fe-O	0.3(3)	2.02(4)	-5.0(6)	0.0083(5)
0.6	Fe-N	3.1(2)	1.96(3)	-13.4(4)	0.0096(5)
	Fe-O	0.5(2)	2.02(2)	-5.1(5)	0.0081(6)
0.4	Fe-N	1.7(3)	1.92(2)	-11.7(6)	0.0089(6)
	Fe-O	0.5(2)	2.01(4)	-6.5(5)	0.0086(6)

Response Figure 6. (a) and (b) *In situ* Raman spectra of Fe SAs-MSP tested in 0.1 M KOH and corresponding I_D/I_G values with applied overpotentials.

5. According to this manuscript, the atomically dispersed Fe active sites could be shielded by SCN⁻. However, the Fe SAs-HP after adding KSCN solutions still displayed high activity for ORR. What is the real active site for Fe SAs-HP after adding KSCN solutions? How did the pore structure engineering affect the poisoning of SCN⁻ for Fe active sites?

Answer: We thank the reviewer for valuable discussions on the KSCN shielding experiment. The KSCN shielding experiment in this work was initially carried out by adding 1 mL of 1 M KSCN solution into 100 mL of 0.1 M KOH electrolyte and then immediately testing LSV curves. The limited loss of activity might be caused by insufficient adsorption equilibrium of SCN^- . Thus, active atomically dispersed Fe sites were partially shielded by SCN^- and suffered from limited current and activity loss. We further performed the KSCN shielding experiment and revealed a time-dependent shielding behavior as shown in response Figure 7a. The SCN^- adsorption/desorption equilibrium on active sites was established in 30 min. Fe SAs-HP suffered a loss of limiting current density (from 5.47 to 4.20 mA cm^{-2}) and $E_{1/2}$ (from 0.94 V to 0.88 V) after KSCN shielding. The oxygenated intermediates may compete with SCN^- for adsorption on Fe active centers under working conditions. It is rational since the O_2 would accept electrons from Fe-N-C, while SCN^- donates electrons under high overpotentials. Increasing the electronic charges of Fe-N-C will promote the electron transfer to adsorbate while suppressing back-transfer. (*J. Am. Chem. Soc.* 2023. DOI:10.1021/jacs.3c09193). Thus, the real active site for Fe SAs-HP was still the active Fe centers due to the dynamic adsorption/desorption behavior of SCN^- driven by applied potentials. **Therefore, the active Fe sites were still considered as the active centers after addition of KSCN solutions.**

To access the influence of pore structure on KSCN poisoning behaviors, KSCN shielding experiment of Fe SAs-MCP and Fe SAs-MSP was carried out by immersing as-prepared catalyst in 0.1 M KOH electrolyte containing 1 mM KSCN solutions for 30 min, followed by LSV tests. The detailed limiting current density (J_L), half-wave potential ($E_{1/2}$), kinetic current density at 0.80 V ($J_k @ 0.80 \text{ V}$) before and after KSCN shielding are listed in response table5. Here, the kinetic current density at 0.80 V was used to assess the activity of as-prepared catalysts. **The retention of kinetic current density for Fe SAs-MCP (50.1 %) was the highest among the three catalysts, followed by Fe SAs-HP (39.0 %) and Fe SAs-MSP (14 %), which could be attributed to the greater affinity of SCN^- with mesoporous Fe sites.**

DFT calculations were then performed to evaluate the affinity of SCN^- to porous Fe sites. The adsorption free energy and adsorption models were exhibited in response table 6 and response Fig. 8, respectively. As for single porous Fe sites, mesoporous Fe- N_4 sites showed stronger affinity of SCN^- than single microporous Fe- N_4 sites. The strong adsorption of SCN^- on mesoporous Fe sites would hamper the charge transfer between oxygenated intermediates and active centers. The weak SCN^- adsorption on microporous Fe- N_4 sites therefore lead to the highest kinetic current retention for Fe SAs-MCP (50.1%). Due to the existence of hybrid microporous and mesoporous sites, Fe SAs-HP exhibited medium kinetic current retention (39.0%) between Fe SAs-MCP and Fe SAs-MSP. Moreover, the proximity effect of adjacent microporous Fe- N_4 sites would also greatly influence the adsorption behavior of mesoporous Fe- N_4 sites, which lead to great loss of kinetic current density from 115.0 mA cm^{-2} to 44.9 mA cm^{-2} . The stable adsorbed SCN^- on active mesoporous Fe- N_4 sites would block the charge and mass transfer to oxygenated intermediates. The microporous Fe- N_4 sites of Fe SAs-HP during KSCN shielding experiment would afford the most ORR activity. **We have replaced supplementary Fig. 10 with Response Fig. 7b.**

Response Figure 7. (a) KSCN shielding experiment of Fe SAs-HP at different time intervals. (b) ORR polarization curves of porous Fe single before and after KSCN shielding experiment.

Response Table 5. ORR performance of as-prepared catalysts before and after KSCN shielding experiment.

Catalyst	$E_{1/2}$ (V_{RHE})	$E_{1/2}$ - KSCN (V_{RHE})	J_L (mA cm^{-2})	J_L - KSCN (mA cm^{-2})	$J_K@$ 0.80 V (mA cm^{-2})	$J_K@$ 0.8V- KSCN (mA cm^{-2})	$J_K@0.80$ V retention
Fe SAs- HP	0.94	0.88	5.47	4.20	115.0	44.9	39.0 %
Fe SAs- MCP	0.86	0.84	5.39	3.73	17.5	8.8	50.1 %
Fe SAs- MSP	0.84	0.80	4.67	4.16	24.3	3.4	14 %

Response Table 6. The adsorption free energy of SCN^- on porous Fe sites.

Active centers	Free energy (E_{ads} , eV)
Fe SAs-HP@MSP	-1.72
Fe SAs-HP@MCP	-1.34
Fe SAs-MCP	-1.39
Fe SAs-MSP	-1.52

Response Figure 8. The simulated SCN^- adsorption models on porous Fe sites. (a) Fe SAs-MCP. (b) Fe SAs-MSP. (c) Fe SAs-HP@MCP. (d) Fe SAs-HP@MSP.

REVIEWERS' COMMENTS

Reviewer #1 (Remarks to the Author):

Thanks for addressing all my questions clearly.

Reviewer #3 (Remarks to the Author):

I have read this revised manuscript carefully. The authors have given sufficient reply to the related questions. So I suggest that this manuscript can be accepted.

Point-by-point response to the referees' comments

We sincerely thank the referee for the careful review on our manuscript (NCOMMS-23-44547A) and the valuable comments and suggestions, which helps to improve our manuscript. We also would like to thank the editor for giving us the opportunity to revise our manuscript. The point-by-point responses are presented below.

Comments by Referees:

Referees #1:

Comments: *Thanks for addressing all my questions clearly.*

Response:

We sincerely appreciate the referee for all the constructive comments and suggestions, which really help to improve our manuscript. We are also very pleased to know the referee is satisfied with our previous response and revision. We again appreciate the referee's recommendation of acceptance and helpful comments in the reviewing process.

Referees #3:

Comments: *I have read this revised manuscript carefully. The authors have given sufficient reply to the related questions. So I suggest that this manuscript can be accepted.*

Response:

We sincerely appreciate the referee for your valuable comments and suggestions. We are also very pleased to know you are satisfied with our previous response and revision. Thank you very much for agreeing to recommend our manuscript.